# Cretaceous-Paleocene extension at the southwest continental margin of India and opening of the Laccadive Basin: Constraints from geophysical data

Mathews George Gilbert[1], Parakkal Unnikrishnan[1], and Munukutla Radhakrishna[1]

[1]Department of Earth Sciences, Indian Institute of Technology Bombay, Powai, Mumbai 400076, India.

**Correspondence:** Mathews George Gilbert (georgegilbertm@gmail.com)

**Abstract.** Previous geophysical investigations of the Western Continental Margin of India (WCMI) confirm the two-phase breakup history of the margin with the first breakup taking place between India and Madagascar that created the Mascarene Basin in the Late Cretaceous and the second breakup event in early Paleocene with Seychelles separating from India. Despite numerous geoscientific studies along the WCMI, the opening of the Laccadive Basin, situated along the southern part of the margin, remains poorly constrained. In this study, we evaluate the multi-channel seismic reflection and gravity anomalies at the margin to identify the early rift signatures in conjunction with the magnetic anomaly identifications in the Mascarene Basin. The analysis led to the identification of two trends of extensional structures, a NNW-SSE oriented structure over the Laccadive Ridge north of Tellicherry Arch, interpreted to result from ENE-WSW extension, and a SSW-NNE oriented structure in the Laccadive Basin region towards the south, interpreted to result from NW-SE extension. Previous plate reconstruction models of the Mascarene Basin using marine magnetic lineations suggest that the ENE-WSW extension observed over the Laccadive Ridge could be related to the India-Madagascar separation. We associate the pattern of sediment deposition and the presence of a Paleocene trap volcanics, linked with the NW-SE grabens observed in the Laccadive Basin region, to the extension between the Laccadive Ridge and the West coast of India after the separation of Madagascar from India. We further propose that the anti-clockwise rotation of India and the passage of the Réunion plume have facilitated the opening of the Laccadive Basin.

## 1 Introduction

The breakup of Gondwanaland into Eastern and Western Gondwana during the Early Jurassic period initiated the formation of western Indian Ocean (fig. 1A-B). The subsequent breakup of Madagascar and the Seychelles from India, and the seafloor spreading along the Carlsberg and Central Indian ridges culminated into the development of the present-day northwest Indian Ocean. The Laccadive Ridge, the Maldives ridge, the Chagos Bank, the Saya-de-Malha Bank, the Nazreth Bank, the Mauritius Island and the Reunion Island are the major topographic features present in the northwest Indian Ocean (fig. 1A), and some of these are believed to be micro-continents (Torsvik et al., 2013).

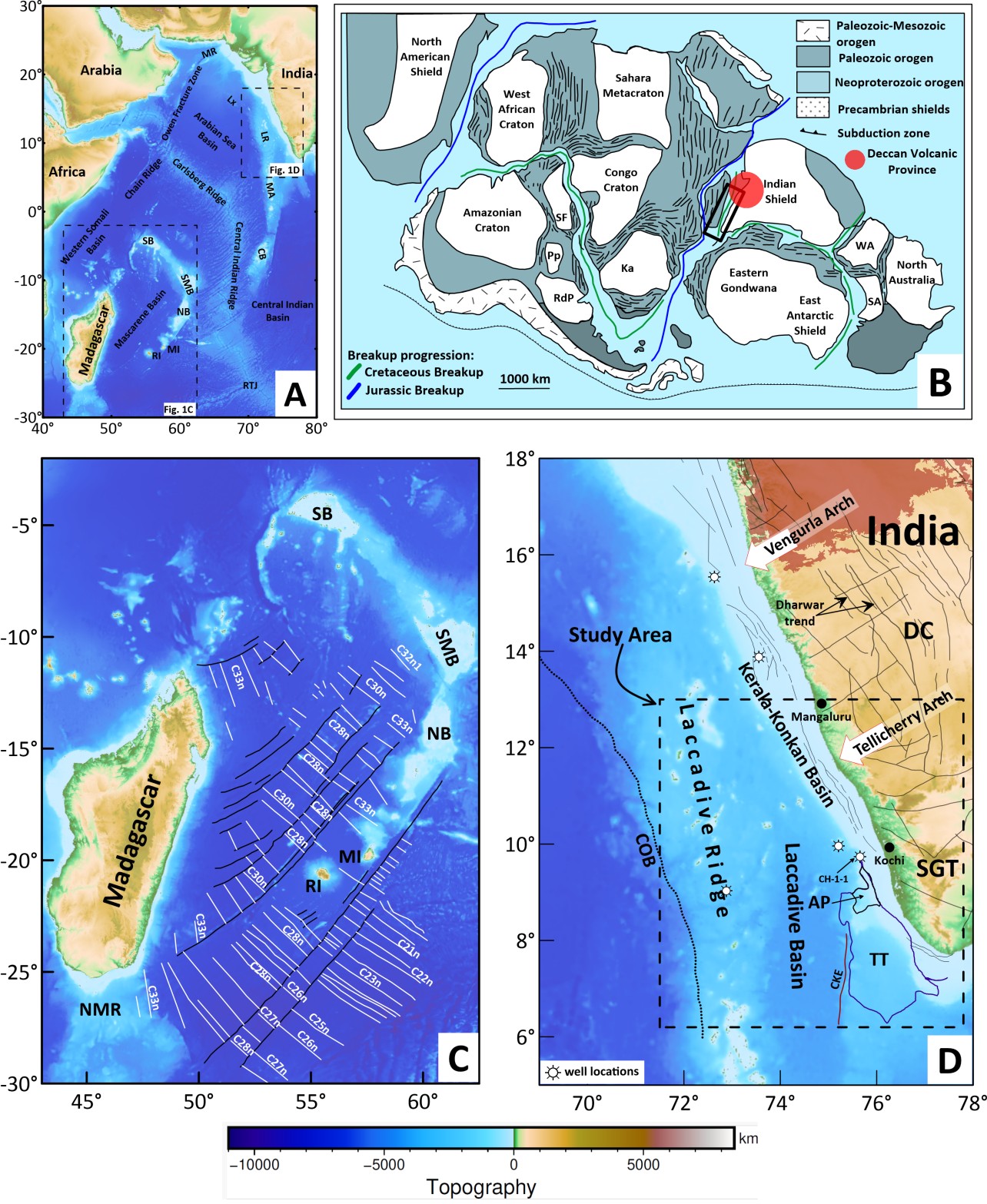

**Figure 1.** A) Regional tectonic map of the northwestern Indian Ocean with satellite-derived seafloor topography (Smith and Sandwell, 1997). B) Map showing the position of India relative to Madagascar and Deccan Volcanic Province in late Paleozoic fit (Lovecchio et al., 2020, modified after). The area of interest is marked in black rectangle. C) Tectonic map of Madagascar and Mascarene basin showing mapped seafloor spreading type magnetic lineations in solid white coloured lines (Bhattacharya and Yatheesh, 2015; Shuhail et al., 2018, and references therein). Solid black lines represent the mapped fracture zones or pseudo-faults. D) Regional tectonic map of the western continental margin of India (Smith and Sandwell, 1997). Black solid lines in the offshore region represent shear zones and faults. The location of the maps in panels C and D are shown in panel A. SB: Seychelles Bank; SMB: Saya-de Malha Bank; NB: Nazarat Bank; MI; Mauritius Island; RI: Reunion Island; RTJ: Rodrigues Triple Junction; LR: Laccadive Ridge; MA; Maldives ; CB: Chagos Bank; MR: Murray Ridge; Lx: Laxmi Ridge; RdP: Río de la Plata; Ka: Kalahari; Pp: Paraná-Panema; SF: Sao Francisco; WA: West Australia; SA: South Australia; NMR: Northern Madagascar Ridge; DC: Dharwar Craton; SGT: Southern Granulite Terrain; LB: Laccadive Basin; AP: Alleppey Platform; TT Trivandrum Terrace; CKE: Chain-Kairali Escarpment; COB: Continent-Ocean transition.

The Western Continental Margin of India (WCMI) formed through the breakup and separation of India and Madagascar in the Late Cretaceous (Storey et al., 1995; Pande et al., 2001). This breakup event resulted in the formation of the Mascarene

Basin (fig. 1A & 1C), the details of which are recorded in the magnetic anomalies in the basin (fig. 1C). The northern part of the margin then experienced another breakup event when the Seychelles block separated from the combined Laxmi Ridge and India in the early Paleocene time. This second breakup event is well studied with the pre-drift juxtaposition of the continental blocks fairly well established from the magnetic anomaly identifications and geochronology (Collier et al., 2008; Chaubey et al., 2002; Ganerød et al., 2011; Shellnutt et al., 2015, 2017). The southern part of the margin is considered to be conjugate

with the eastern Madagascar margin (Katz and Premoli, 1979) based on the continuity of the major shear zones and coastlines matched at 1000 m isobath. However, recent close-fit reconstruction models have incorporated the continental fragments like the Laccadive Ridge (Bhattacharya and Yatheesh, 2015) or Mauritia (comprising of Mauritius, the Southern Mascarene Plateau, the Laccadive Ridge and the Chagos Bank) (Torsvik et al., 2013) between India and Madagascar in the India-Madagascar pre-drift scenario, and suggested a breakup timing of around 83 Ma.

The Laccadive Ridge is the bathymetric high feature in the southwest offshore margin and lies parallel to the west coast of India (fig. 1D). The Laccadive Basin lies between the Laccadive Ridge and the continental shelf south of Tellicherry Arch (fig. 1D). The southwest margin of India was affected by Réunion plume volcanism towards the end of the Cretaceous (Singh and Lal, 1993) as revealed by the presence of a wide-spread layer of trap volcanics below the Tertiary sediment cover at the margin (Singh et al., 2007; Singh and Lal, 1993).

Most of the drilled-wells along this margin were terminated in the Late Cretaceous to Early Paleocene trap layer and none till date have encountered the crystalline basement. By contrast, the CH-1-1 well located within the shelf (fig. 1D) penetrated through the Paleocene trap, encountered Santonian formations and terminated in the Late Cretaceous felsic volcanics (Singh and Lal, 1993). One of the key questions that have not been resolved concerns the absence of Late Cretaceous sediments in the Laccadive basin as a whole: what caused this more than 20 Myr gap in the sedimentary record between India-Madagascar

breakup at 83 Ma and the oldest Paleogene sediments? This long time gap indicates either the presence of older sediments below the Paleocene trap layer or the opening of the Laccadive basin subsequent to India-Madagascar separation. Further, the presence of the continental fragments between India and Madagascar (Bhattacharya and Yatheesh, 2015; Torsvik et al.,

2013) makes for a complex geodynamic setting, considering how this separation took place, and therefore provides some insights into the impact of pre-existing lithospheric inheritance. Hence, examining the development of the Laccadive Basin will provide important constraints on the early breakup evolution of the WCMI.

In this study, we made a correlative analysis of the multi-channel seismic data with the residual gravity anomalies which provided evidence of a major extensional event that occurred at the southwestern part of the margin that is not related to the India-Madagascar breakup. The time-stamping of this major extensional event provides important constraints on the evolution of the WCMI and help to build improved plate tectonic reconstruction models.

## 2    Description of tectonic elements of the study area

The study area lies south of Tellicherry Arch and contains the southern part of the Laccadive Ridge and the Laccadive Basin (fig. 1D). The major geomorphic features present in the study area from west to east are the Laccadive Ridge, the Laccadive Basin, the Alleppey Platform and Trivandrum Terrace (together called the Alleppey-Trivandrum Terrace Complex) and the continental shelf (fig. 1D). The Alleppey-Trivandrum Terrace Complex is characterized by horst-graben structures and bounded in the west by the Chain-Kairali Escarpment (CKE) feature (Yatheesh et al., 2006, 2013; Nathaniel, 2013) as revealed in seismic sections (See fig. S1). Numerous seamounts/guyots/knolls are present in the region between the Laccadive Ridge and the continental shelf within the Laccadive Basin (See fig. S2) (Bijesh et al., 2018). In addition, the entire region is characterized by several intrusive structures within the Tertiary sediments (Unnikrishnan et al., 2023). The southern part of the Cannanore Rift system (CRS) identified over the Laccadive Ridge (fig. 2) from the Director General of Hydrocarbon (source: DGH2024) lies within the study area.

## 3    Data and Methods

In this study, we used the satellite-derived free-air gravity (fig. 2A) (Sandwell et al., 2014) and bathymetry from General Bathymetric Chart of the Oceans (fig. 1D) (GEBCO, 2020) for comparative analysis with multi-channel seismic reflection profiles (figs. 2B-5). The large volume of industry seismic reflection data (see supplementary fig S3) at this margin provided information on the sediment thickness above the Paleocene trap layer and various horizons within the post-Paleocene sediments (Unnikrishnan et al., 2023). We also gathered a few published seismic sections (Nathaniel, 2013; Yatheesh et al., 2013) within the study area (see supplementary fig S1).

The gravity effects of the water layer and the sediments are removed from the satellite-derived free-air anomaly data to obtain crustal Bouguer anomalies. High-resolution sediment thickness derived from two-way travel time (TWT) maps is used to calculate the gravity effect of the sediments. The TWT maps are available for three different times: the early Paleocene, early Eocene, early Miocene (Unnikrishnan et al., 2023). These maps were converted to depth with respective interval velocities (after Unnikrishnan et al., 2023) and the total sediment layer is used to calculate the gravity effect of sediments. The densities of 2300 kg/m3 for sediments and 1030 kg/m3 for the water column are used respectively . For the crustal rocks, an average density

of 2800 kg/m3 was considered as the study area lies within the extended continental crust (Unnikrishnan et al., 2023). A 10-
200 km wavelength band-pass filter was applied to the crustal Bouguer anomaly map to highlight the crustal heterogeneities,
and the first vertical derivative (FVD) as well as the total horizontal derivative (THD) of the band-pass filtered map was
prepared to identify shallow structural features. The location of the identified features such as rifts and volcanic intrusives in the
seismic sections are then transferred on to these gravity anomaly maps in order to understand their continuity. Subsequently,
the sediment isochron maps were interpreted to understand the timing of opening of the Laccadive basin with inputs from
interpreted seismic sections. Scientifically derived colour maps are used to prepare maps (Crameri et al., 2020).

## 4 Results

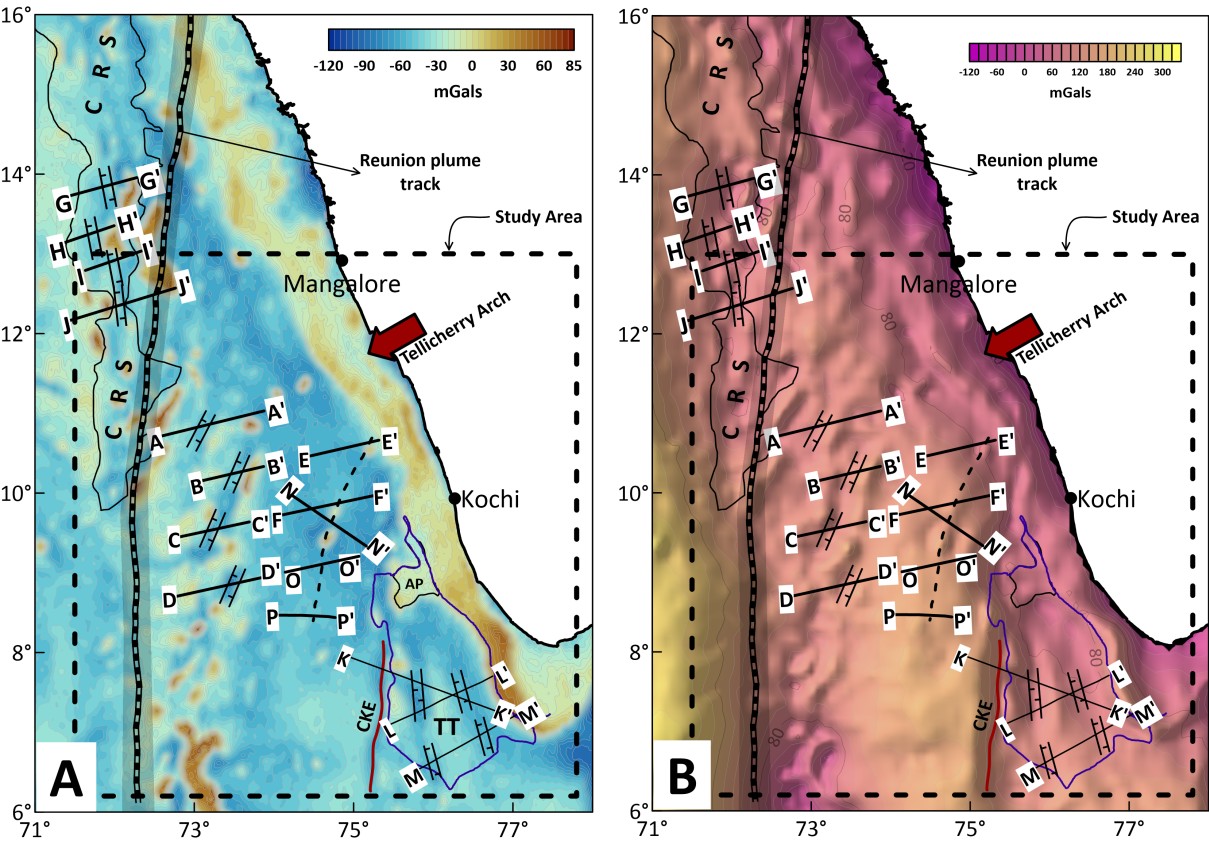

**Figure 2.** A) Satellite-derived free-air gravity anomaly map (Sandwell et al., 2014) B) Crustal Bouguer anomaly map. The location and orientation of identified extensional features, grabens and intrusives are marked. Black solid lines represent the location of the profiles. Interpreted seismic sections are shown in fig. 5. The thick broken black line in the centre of the basin represents the identified volcanic ridge. (refer to fig. 5 & S1 for seismic sections). The location of the maps is shown in fig. 1A. CRS represents the Cannanore Rift System as identified by DGH2024. CKE: Chain-Kairali Escarpment; AP: Alleppey Platform; TT: Trivandrum Terrace.

## 4.1 Sediment Isochron map analysis

The isochron maps in the study area (fig. 3) show that for the early Paleocene to early Eocene time interval, significant sediment deposition occurred parallel to the coast with much less sedimentation in the Laccadive Basin. During this period,

maximum deposition took place in the area between Tellicherry Arch and off Kochi (marked sediment patch in fig. 3A) with a minor sediment channel extending into the basin (marked sediment channel in fig. 5A). During the Early Eocene to Early Miocene time interval, deposition within the sediment patch was almost absent, whereas, significant deposition is observed in the northern part of the Laccadive Basin on either side of the identified volcanic intrusives (discussed below) (fig. 5B). From Early Miocene to recent times, sedimentation has been uniform in the Laccadive Basin (fig. 3C). Sediment deposition along

the coast remained high throughout the time intervals (fig. 3A-C). The total sediment deposition pattern shows that most of the sediments were accommodated parallel to the coast and towards the south there is an axis of high sediment deposition into the Laccadive Basin (fig. 3D).

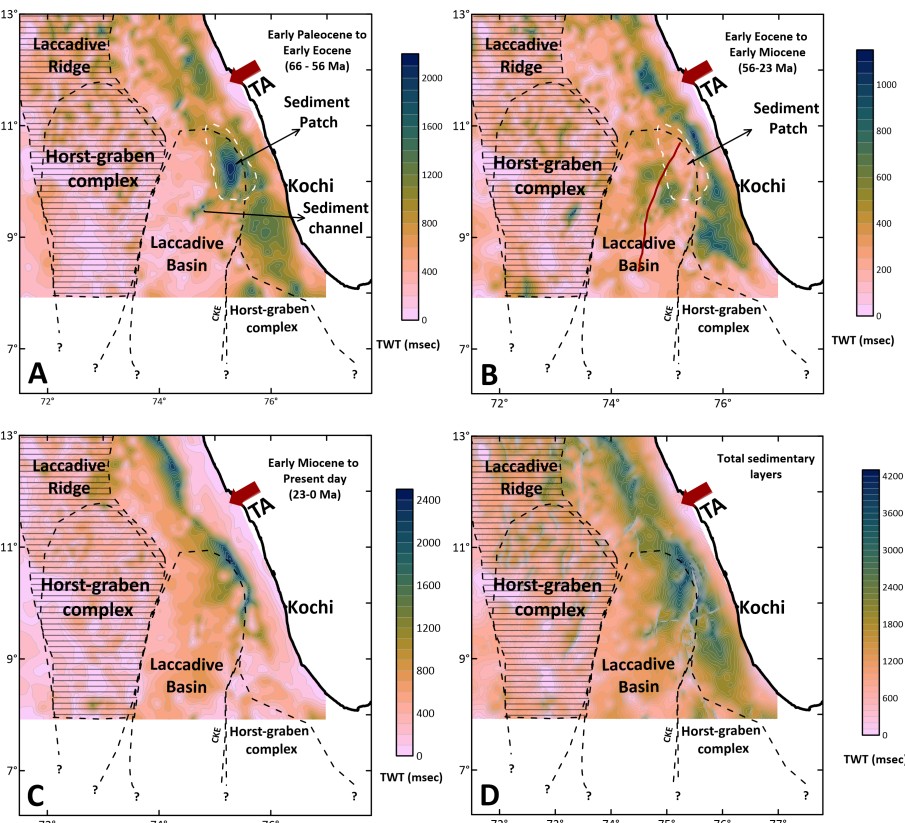

**Figure 3.** Isochron maps from Unnikrishnan et al. (2023) for selected time intervals: A) Early Paleocene to Early Eocene; B) Early Eocene to Early Miocene; C) Early Miocene to Present day; and D) Total sedimentary layers. The brown line in panel B represents trend of the identified volcanic Ridge. (Refer to text for detailed explanation and interpretation). TA: Tellicherry Arch; CKE: Chain-Kairali Escarpment.

## 4.2 Gravity anomaly mapping

The crustal Bouguer anomaly map reveal two structural trends: NNW-SSE orientation in the area north of the Tellicherry Arch and NNE-SSW orientation south of Tellicherry Arch (see fig. 2B). The band-pass filtered map of crustal Bouguer anomaly and its first vertical derivative and total horizontal derivative maps in the study area further enhanced the structural features observed in the crustal Bouguer anomaly map (fig. 4).

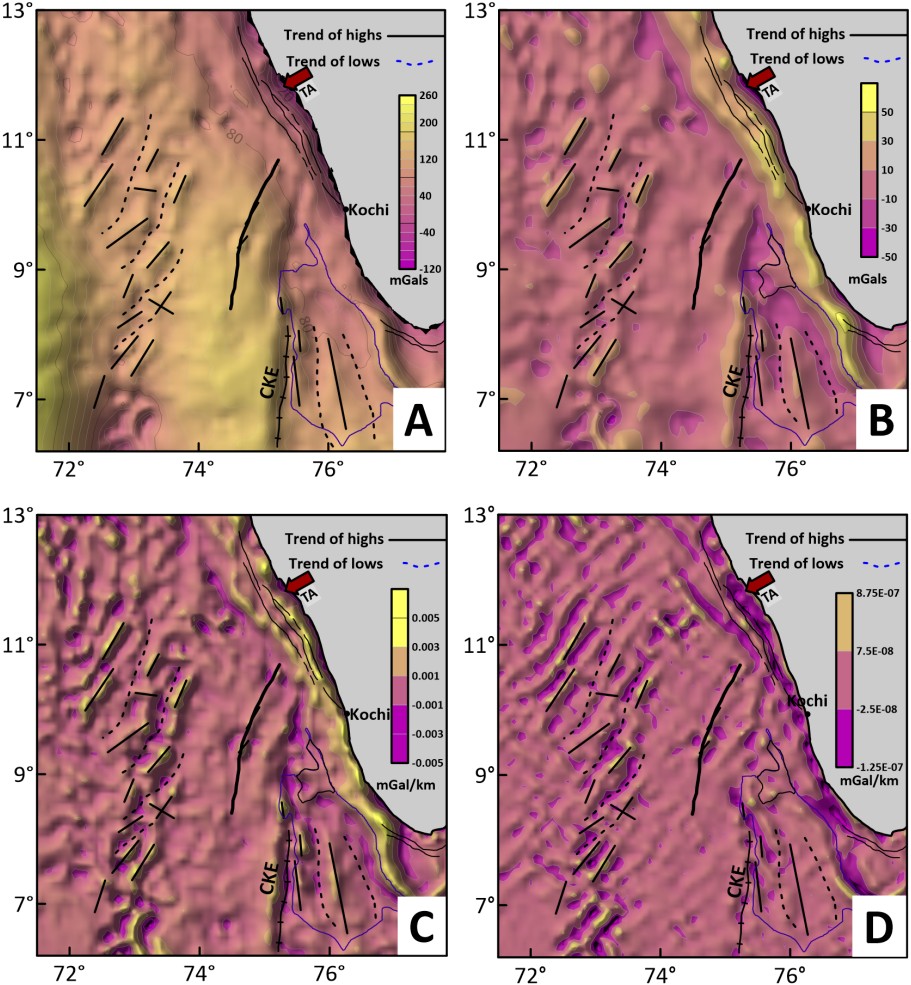

**Figure 4.** A) Crustal Bouguer anomaly, B) Band-pass filtered crustal Bouguer anomaly, C) First vertical derivative (FVD) of band-pass filtered crustal Bouguer anomaly, D) Total horizontal derivative (THD) of band-pass filtered crustal Bouguer anomaly. The black lines show the structural highs and the black dotted lines show the continuity of rift basins identified. The black coloured solid ticked line represents the Chain-Kairali escarpment (CKE). The black thick solid line along the centre of the basin represents the identified volcanic ridge. The thin black lines close to the coast represents major faults identified on the continental shelf Singh and Lal (1993). TA: Tellicherry Arch.

## 4.3 Seismic interpretation

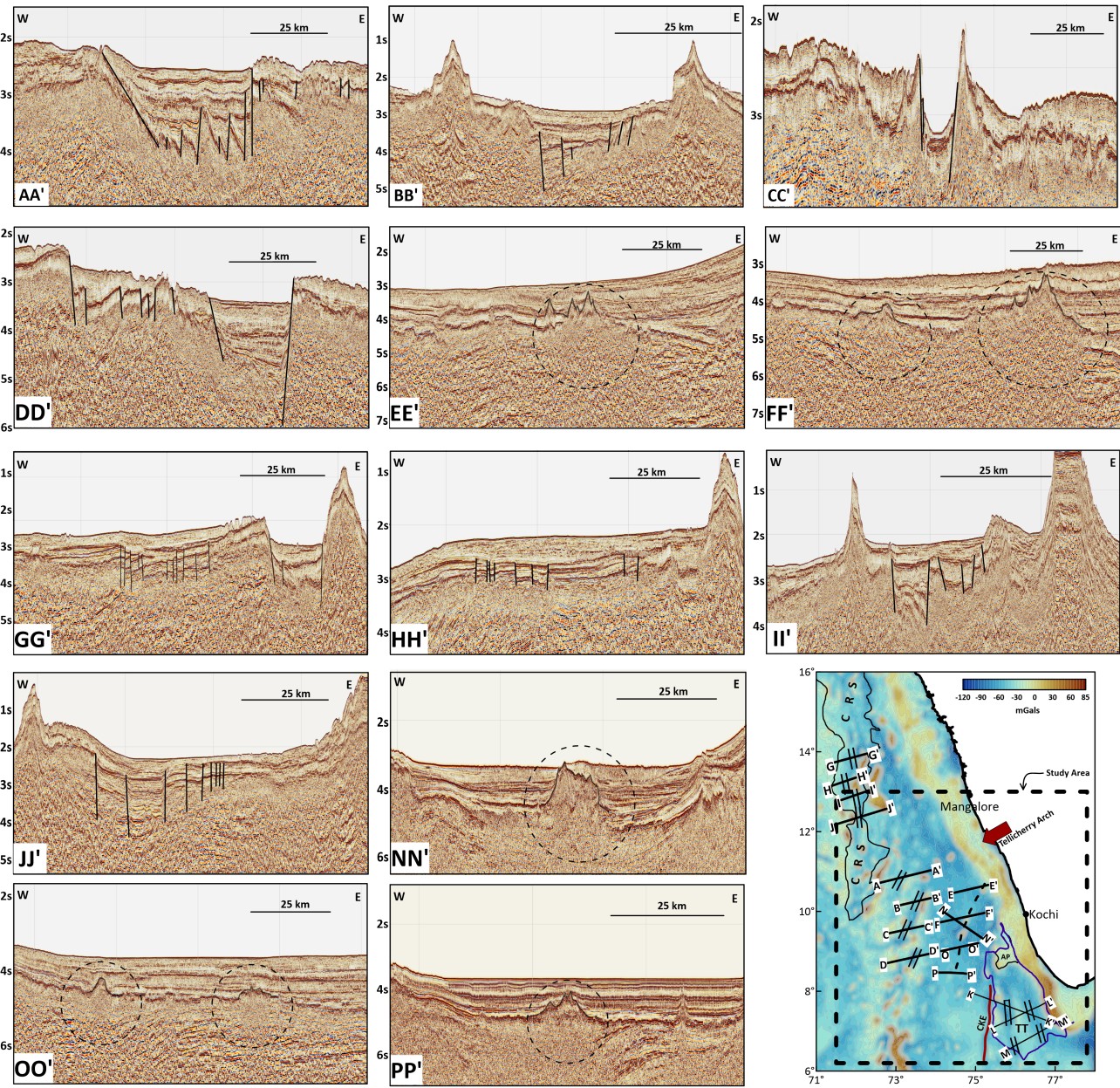

**Figure 5.** Figure showing interpreted seismic sections. The faults are marked and the intrusives are shown in dotted circles. The inset free-air anomaly map on the right bottom corner shows that location of seismic profiles. The time is given two-way travel times (TWT). TWT is the elapsed time for a seismic wave to travel from its source to a given reflector and return to a receiver at the Earth's surface.

We present thirteen interpreted seismic sections (fig. 5) which reveal several rift-related horst-graben structures (extensional features) at the margin. We correlated these structures with the gravity anomaly trends and noticed that the grabens are oriented NNW-SSE in the area north of the Tellicherry Arch, whereas the grabens are oriented NNE-SSW south of Tellicherry Arch (see fig. 2, 3 & fig. 5). These identified extensional structures show low gravity anomalies, the continuity of which can be traced as gravity lows surrounded by highs in the anomaly maps (fig. 2A-B & fig. 4A-D). This is particularly prominent within the study area (south of Tellicherry Arch). The seismic sections also reveal series of volcanic intrusives in the centre of the Laccadive Basin which is seen in the gravity anomaly map also as a broken chain of highs (fig. 2, 3 & 4). The NNW-SSE oriented grabens over the Laccadive Ridge north of Tellicherry Arch are part of the CRS mapped by the DGH.

## 5   Discussion

This study identifies two major structural trends along the southwest margin, the first one, the NNW-SSE oriented grabens over the Laccadive Ridge north of Tellicherry Arch, and the second, the NNE-SSW graben system in the Laccadive Basin area south of Tellicherry Arch (fig. 2). The NNW-SSE rifts observed over the Laccadive Ridge conforms with the orientation of the Dharwar structural trend which is a dominant structural trend in the onshore region north of Tellicherry Arch (fig. 1D). Also, these are part of the CRS that is characterized by complex block-like basement structures comprising of grabens, half-grabens and faults (source: DGH2024) along the eastern flank of the Laccadive Ridge and extends between $17^0$N-$9.5^0$N parallel to the west coast of India (fig. 2A-B). Previous studies (Kolla and Coumes, 1990; Singh and Lal, 1993) also inferred the continuation of onshore Dharwar trend in the offshore shelf region and beyond. As mentioned earlier, India-Madagascar breakup took place at 83 Ma and resulted in the opening of the Mascarene Basin. Magnetic anomaly identifications together with plate-tectonic reconstruction studies (Shuhail et al., 2018) reveal that during the initial period (83-79 Ma), the spreading in the Mascarene Basin was E-W (fig. 1C), and this initial spreading regime matches well with the NNW-SSE oriented rift system observed over the Laccadive Ridge (fig. 2). Hence, we infer that the NNW-SSE oriented grabens had formed during the early breakup evolution of the margin between India and Madagascar.

Coming to the margin south of Tellicherry Arch, it is relevant to understand the spreading regime in the Mascarene Basin. Analysis of magnetic anomaly identifications (Bhattacharya and Yatheesh, 2015; Shuhail et al., 2018) reveal that the spreading was more active in the southern part of the basin and also characterized by longer transform faults between the spreading segments. If reconstructed backward to this time (73 Ma reconstruction Shuhail et al. (2018)), the identified grabens south of Tellicherry Arch follow the orientation of these transform faults. Plate tectonic reconstruction efforts by Shuhail et al. (2018) indicate that the spreading in the Mascarene Basin was connected to the study area by long transform faults such as the CKE. Hence, we interpret that the SSW-NNE fault system developed during this time when the study area was in close proximity to the Mascarene Basin spreading center (fig. 7 Stage II) due to movement along the transform faults, whereas such large transforms with active spreading was missing in the northern part of the Mascarene Basin which is conjugate with the Laccadive Ridge region north of Tellicherry Arch. Subsequently, spreading ceased in the Mascarene Basin and the basin was transferred to the African plate by 57 Ma. Later, these faults opened and extension continued contemporaneous with India's

anti-clockwise rotation which facilitated the opening of Laccadive Basin as indicated by the sediment isochron maps (fig. 3A-C).

## 5.1 Opening of the Laccadive Basin

As explained in section 4.1 the sediment deposition in the basin is interpreted from the perspective of the creation of accommodation space and sediment supply. The high-resolution time-structure maps (fig. 3A-D) provide insights on the timing of opening of the Laccadive Basin. These maps clearly reveal significant sediment deposition along the coast parallel grabens within the shelfal part of the margin in all time periods. Further, during the Paleocene-Eocene period, sediment deposition was very significant on the northern fringe of the Laccadive Basin (sediment patch in fig. 3A) with negligible sediments elsewhere in the basin. During Eocene to Miocene the sediment deposition shifted further offshore into the Laccadive Basin (fig. 3B). The development of the volcanic ridge within the basin also occurred during this time since it is observed that the sediments are deposited on either side of the location of the ridge. Further the seismic sections show sediments onlapping to the identified volcanic ridge (fig. 5 sections NN' & PP'). The comparative analysis of fig. 3A & 3B indicates that the basin opened some-time after the early Eocene as a result of which accommodation space was created and all the incoming sediments migrated southward into the basin. A small channel of sediment deposition into the Laccadive Basin towards the southwest of the sediment patch (marked as sediment channel in fig. 3A) may represent the initial stage of opening of the basin. During the Miocene to recent period (fig. 3C), the sediment deposition is more or less uniform throughout the basin. The distal part of the Laccadive basin towards west has relatively less sediment deposition which may be due to the area's location far from any sediment supply.

Unnikrishnan et al. (2018) identified the Alleppey platform as a continental fragment and inferred its development during the Oligocene-Miocene period. Alleppey platform is located adjacent to the Laccadive Basin and hence the development of the basin and the platform could be related. The timing of the development of the Alleppey platform given by Unnikrishnan et al. (2018) closely agrees with the inferred timing of the opening of the Laccadive Basin from this study.

## 5.2 Distribution of Bathymetry highs and intrusives

A striking feature along this margin is the presence of many intrusives and bathymetric highs (see fig. S2) observed in the seismic and bathymetry data, respectively (Unnikrishnan et al., 2023; Bijesh et al., 2018). These features have very clear expressions on the gravity image of the area (figs. 2 & 3A-D). The intrusives and bathymetric highs in the study area, south of Tellicherry Arch (fig. S2) appear to be elongated roughly parallel to the trend of the Laccadive Basin. In the centre of the Laccadive Basin, we noticed a series of volcanic mounds with a trend almost parallel to the CKE (fig. 2), which are clearly expressed in the seismic sections (fig. 5). The observed trend correlates well with the crustal Bouguer anomaly map (fig. 4A) as well as the depth to basement map prepared by adding the sediment thickness to bathymetry (fig. 6A). This trend divides the Laccadive Basin into eastern and western basins. The composite tectonic map of the study area is shown in fig. 6C.

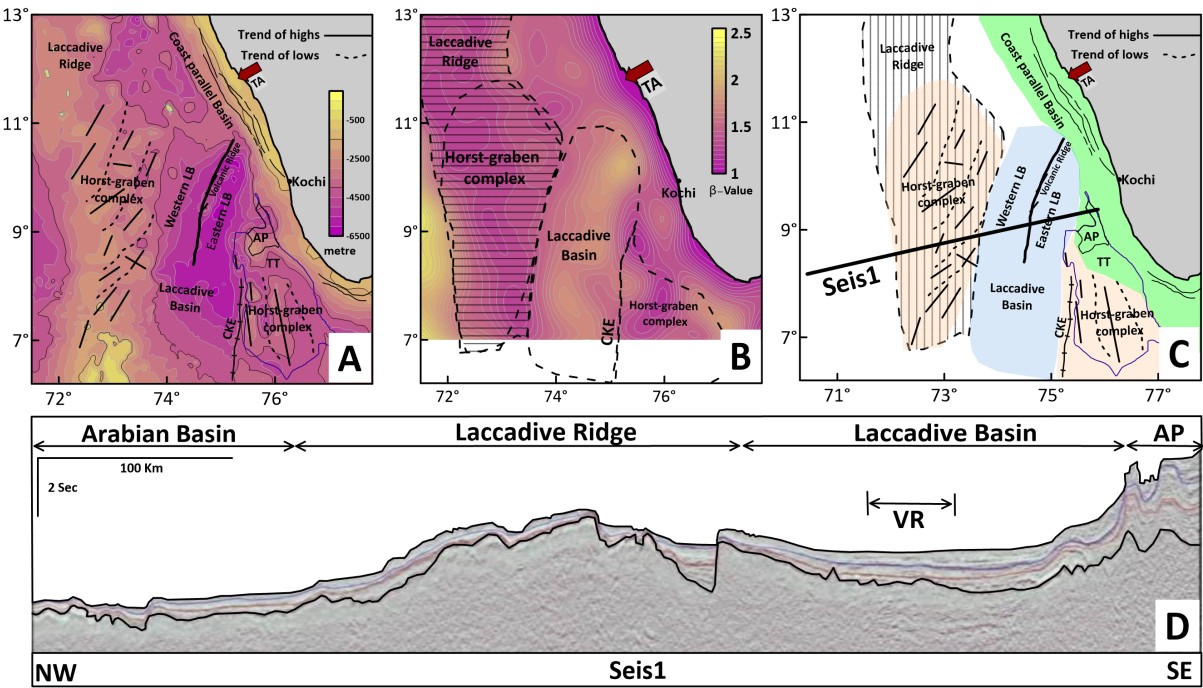

**Figure 6.** A) Depth to the basement map with all structures identified in the study, B) $\beta$-value map redrawn from Unnikrishnan et al. (2023), C) Proposed tectonic map of the study area. Bottom panel shows a seismic section (Seis1), showing the general characteristic of the Laccadive Basin area. The seismic section is from Unnikrishnan (2018). The black lines show the structural highs and the black dotted lines show the continuity of rift basins identified. The black coloured solid ticked line represents the Chain-Kairali escarpment (CKE). The thick solid black line represents the identified volcanic ridge. The thin black lines close to the coast in A & C represents major faults identified on the continental shelf (Singh and Lal, 1993). TA: Tellicherry Arch; AP: Alleppey platform; TT: Trivandrum Terrace; CKE: Chain-Kairali Escarpment. AP: Alleppey platform; VR: Volcanic Ridge; LB: Laccadive Basin.

Further, the $\beta$-value (crustal stretching factor) map (fig. 6B) calculated by Unnikrishnan et al. (2023) clearly reveals the exten-
sional trend in the study area. The high $\beta$-values in the centre of the Laccadive Basin indicate maximum thinning and conforms
with our observation in this study.

### 5.3  Evolutionary model

Existing plate tectonic reconstruction models (Bhattacharya and Yatheesh, 2015; Shuhail et al., 2018), the pre-existing struc-
tural trends along WCMI, India and Madagascar (Subrahmanyam et al., 1994; Kolla and Coumes, 1990; Bhattacharya and
Yatheesh, 2015), and the results from this study have been used to build a schematic evolutionary model for the opening of the
Laccadive Basin at the southwest margin of India. Additionally, we also looked at analog and numerical modelling studies that
explain extensional deformation in rifts and rifted margin systems (Péron-Pinvidic and Manatschal, 2010; Zwaan et al., 2021;

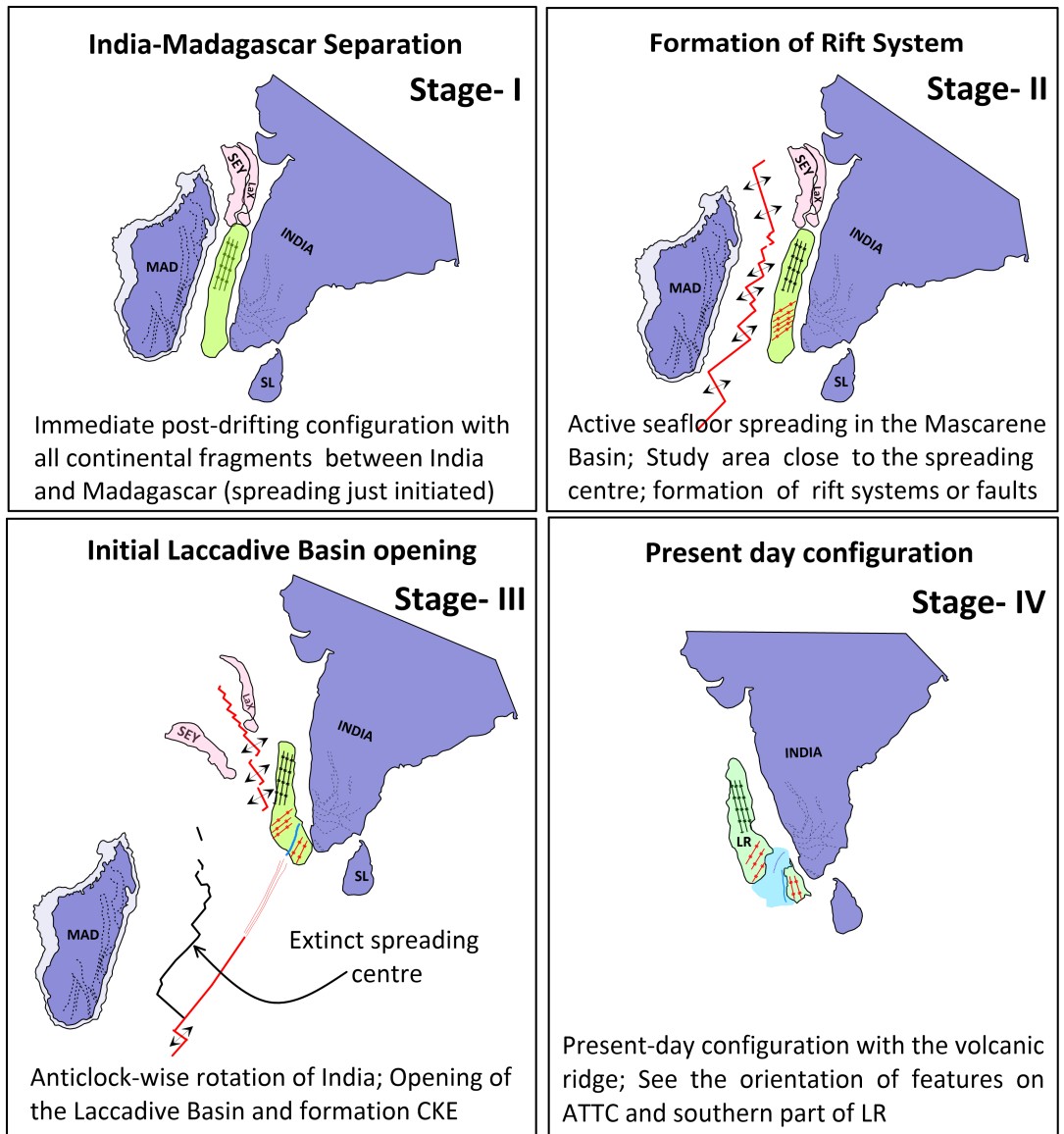

**Figure 7.** Map showing the evolution of the region in four stages. Stage I: The pre-rift juxtaposition of the continental fragments. Note that the Laccadive Ridge is larger since it incorporates all the fragments that are littered between India and Madagascar. Stage II: The formation of the faults system or the rifts system due to the influence of spreading in the Mascarene Basin. Stage III: The opening of the basin with CKE + ATTC and India moving away from the Laccadive Ridge. See how the orientation of the extensional feature's changes with the anticlockwise movement of India. Stage IV: The present-day configuration of the margin with all extensional features and the volcanic ridge. LaX: Laxmi Ridge; SEY: Seychelles; MAD: Madagascar; SL: Sri Lanka; LR: Laccadive Ridge; ATTC: Alleppey-Trivandrum Terrace Complex; CKE: Chain-Kairali Escarpment.

Bonini et al., 1997; Henza et al., 2010). The locations of India and Madagascar in various stages (I-IV) (fig. 7) were adopted from the reconstruction model of Shuhail et al. (2018). The salient aspects of each stage is given below:

### 5.3.1 Stage I

Stage I (fig. 7-Stage I) shows India and Madagascar along with all the continental fragments in between them and a large number of shear zones onshore for both India and Madagascar (Bhattacharya and Yatheesh, 2015, and references therein) immediately after the spreading started at 83 Ma. These shear zones were earlier used as piercing points to find the relative position of India and Madagascar in the matching of coastlines (Katz and Premoli, 1979; Subrahmanyam and Chand, 2006). Besides, studies along the WCMI have shown the extension of onshore structural trends into the offshore region (Subrahmanyam et al., 1994; Kolla and Coumes, 1990) and studies show that structural inheritance plays a role during rifting and breakup (Péron-Pinvidic and Manatschal, 2010; Zwaan et al., 2021; Bonini et al., 1997; Henza et al., 2010). Dating of volcanics in Madagascar yields an age of 88 Ma (Storey et al., 1995) and magnetic anomalies indicate spreading started around 83 Ma (Shuhail et al., 2018). Hence, we infer that during this extensional span, CRS formed on the Laccadive Ridge following the Precambrian Dharwarian trend which is dominant north of Tellicherry Arch.

### 5.3.2 Stage II

Stage II (fig. 7-Stage II) shows the proximity of Laccadive Ridge to the spreading centre in the Mascarene basin. As discussed in stage I, the lithosphere between India and Madagascar had zones of weakness (as evidenced by the presence of inherited structures) and as a result, when the area was proximal to the spreading centre, a number of parallel trans-tensional faults may have formed on the Laccadive Ridge south of Tellicherry Arch. Even though the spreading was happening along the entire basin, the spreading was very active in the southern part of the Mascarene basin, and also characterized by long transform faults (Shuhail et al., 2018). As discussed in section 5.2 distribution of bathymetric highs and intrusives south of Tellicherry Arch provide some evidence for this. It is very likely that, later when the Réunion plume passed over the area, magma may have migrated through the faults formed during this stage giving rise to the preferred orientation of intrusive and bathymetric features in this area. It is worthwhile to note that Bijesh et al. (2018) related the genesis of the bathymetric features to hotspot volcanism.

### 5.3.3 Stage III

Stage III (fig. 7-Stage III) shows the initial stages of opening of the Laccadive basin in the Paleocene. By this time, the entire region was flooded by Deccan volcanics during the passage of Réunion plume. Studies by Patriat and Achache (1984) and Dewey (1989) showed that the Indian plate rotated anticlockwise about $40^0$ since 84 Ma, out of which, it underwent about $25^0$ after the soft collision at 50 Ma (Treloar and Coward, 1991). The passage of the Réunion plume over the area may have weakened the overlying lithosphere and together with India's anti-clockwise rotation and the presence of inherited structural weakness may have led to the reactivation and further extension in the Laccadive Basin. The sediment thickness data during the

Paleocene-Eocene period shows a sediment channel (fig. 3A) extending into the Laccadive basin and an area of high sediment deposition (Sediment patch in fig. 3A) which indicate the initial stages of opening of the basin. fig. 3B shows sediments being deposited on either side of the volcanic ridge in the Laccadive basin. By this time the basin opened and sediments were accommodated in the basin.

### 5.3.4 Stage IV

Stage IV (fig. 7-Stage IV) shows the present-day configuration of the Laccadive Basin. The Alleppey-Trivandrum Terrace Complex remain attached to the Indian continent with CKE forming its western boundary. The centre of the Laccadive Basin experienced maximum crustal thinning and a series of intrusives got emplaced in the crust. The change in the orientation of horst-graben structures in the Alleppey-Trivandrum Terrace Complex from that in the southern part of Laccadive Ridge can be noticed.

## 6 Conclusions

The seismic and gravity data analysis along southern part of WCMI revealed two significantly different orientations of the extensional graben system, one a NNW-SSE oriented rifts over the Laccadive Ridge, north of Tellicherry Arch, and the other NNE-SSW oriented rifts in the Laccadive Basin region south of Tellicherry Arch. The change in extensional direction in the Laccadive Basin along with the isochron maps for different times suggests that the basin opened during the post-Eocene period with maximum extension along the centre of the basin where the volcanic intrusives are emplaced. The lithosphere that existed had zones of weakness and this along with proximity to the Réunion plume and anti-clockwise rotation of India led to complex rifting and evolutionary history for the southern part of the margin.

*Data availability.* The authors do not have permission to share data.

*Author contributions.* MGG – conceptualisation, methodology, validation, formal analysis, writing (original draft and editing), visualisation. MR – conceptualisation, validation, resources, writing ( review and editing), supervision. PU – conceptualisation, writing (review and editing).

*Competing interests.* The authors declare that they have no known competing financial interests or personal relationships that could have appeared to influence the work reported in this paper.

*Acknowledgements.* We would like to thank the Editor, Dr. Frank Zwaan, for the thorough and constructive reviews. We sincerely appreciate all valuable comments and suggestions by Dr. Yatheesh V. and the anonymous reviewer, which helped us improve the manuscript. The work
benefitted from many constructive discussions with Dr. Kondepudi V S S Sai Pattabhiram. This study forms a part of PhD work of MGG at IIT Bombay. MGG thanks the Council of Scientific and Industrial Research (CSIR), New Delhi, India, for the Financial support under Junior Research Fellowship (Grant No. 09/087(1005)/2019-EMR-I). The second author gratefully acknowledges the permission of the Oil and Natural Gas Corporation Limited (ONGC) to use the geoscientific data pertaining to the Kerala-Konkan Basin. The authors acknowledge the financial support from Science & Engineering Research Board ( SERB/CRG/2021/006505).

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
