# Peer review of "Cretaceous-Paleocene extension at the southwest continental margin of India and opening of the Laccadive Basin: Constraints from geophysical data"

_EGUsphere, 2023_

## Referee Comment (RC1)

**Review of the paper**
**(egusphere-2023-1757)**

| | |
|---|---|
| **Authors** | Gilbert M. George, P. Unnikrishnan, M. Radhakrishna |
| **Title** | Evidence of extension at the southwest continental margin of India and opening of the Laccadive Basin |

**OVERALL ASSESSMENT**

The paper deals with a detailed understanding of the southwestern continental margin of India and its adjoining offshore regions based on the rift structures inferred in the Laccadive Plateau region and a curvilinear volcanic trend inferred from the Laccadive Basin, mainly by using the long-offset multichannel seismic reflection profiles. For this, the authors used eight deep-penetrated multichannel seismic reflection sections over the Laccadive Plateau, three seismic section over the Alleppey-Trivandrum Terrace Complex, and two seismic sections over the Laccadive Plateau. The authors derived inferences based on the interpretation of the basement features in terms of rift structures and volcanic intrusives, and evaluated these results with the computed crustal Bouguer anomalies and depth to the basement map.

The results of the present study are based on adequate data and the inferences made by the authors over the Laccadive Plateau and the Alleppey-Trivandrum Terrace Complex are well demonstrated by observations depicted from the seismic sections, complemented by the gravity signatures. All the inferences made by the authors on the opening of the Laccadive Basin based on the isochron map and sedimentation history also appears to be acceptable, however, their structural interpretation on presence of a volcanic intrusive features with a curvilinear trend appears to be too weak to be accepted since this inference is put forward only based on two seismic sections, and such signatures are not clearly visible either in the crustal Bouguer anomaly map, or in the depth to the basement map.

The interpretations provided for the Laccadive Plateau and the Terrace off Trivandrum are quite reasonable, and overall, the paper is well written. The paper falls very well within the standard of the papers published in the EGU journals. My specific comments are given below for the improvement of the paper. I recommend to accept the paper, after incorporating the minor / moderate revisions effecting the following comments provided.

**SPECIFIC COMMENTS**

1) In the whole text: "Reunion" may be corrected as "Réunion" in throughout the manuscript.

2) Lines 10-12: Please modify the sentence in view of the

3) Line 9: "pre-rift" may be modified as "pre-drift" since the age information cannot be derived from magnetic anomalies observed from rift stage crust, but possible only when these magnetic anomalies are formed by seafloor spreading (i.e., drifting"). Therefore, magnetic anomalies can provide only "pre-drift" juxtaposition.

4) Lines 9-10: The detailed mapping of seafloor spreading magnetic anomalies in the conjugate Arabian and eastern Somali basins (spreading between India-Laxmi Ridge block and Seychelles) was published by Chaubey et al. (2002, Geological Society, London, Special Publication 195, pp. 71-85). The same may be quoted here. Although you mentioned geochronology here, the references are missing, please add the same.

5) Figure Caption 1: "MR: Murray Ridge" may be deleted since you have not used this abbreviation in the figure, it is written in expanded form in the figure.

6) Lines 2123: You mentioned *"…… whereas, more recent studies (Torsvik et al. (2013); Bhattacharya Yatheesh (2015) incorporate…."*. This sentence is misleading since Bhattacharya and Chaubey (2015) has not included Mauritius in their model. So, this sentence may be modified as "……whereas, more recent studies incorporate continental fragments like Laccadive Ridge (Bhattacharya and Yatheesh, 2015) or Mauritia, consisting of Mauritius, Southern Mascarene Plateau, Laccadive Plateau and Chagos Bank (Torsvik et al. (2013) between India and Madagascar in the India-Madagascar pre-drift scenario".

7) Line 37: "complicity" may be corrected as "complexity".

8) Line 39: The sentence "……. Laccadive Basin area will shed light on the margin's evolution …." May be modified as ""……. Laccadive Basin area will provide important constraints on the margin's evolution ….".

9) Lines 79-83: It is mentioned that "*Further, a curvilinear trend of volcanic intrusive features is identified in the Centre of the Laccadive Basin parallel to the identified extensional trend. This trend is also observed in the gravity anomaly map as a broken chain of highs*". This inference does not appear to be convincing. First of all, you have only two seismic sections in which intrusives are mapped, therefore, with these two profiles, we can neither interpret the continuity of the features nor its arcuate trend. In addition, I am unable to identify any clear and convincing curvilinear trend from any of the maps presented in Figure 3". Your other interpretations on the ENE-WSW and NW-SE extension on the Laccadive Plateau are convincing as it is clearly observed from the seismic sections. Bringing the inference of Laccadive Basin trend actually dilute the quality of the paper. Therefore, this inference appears to be too weak to be accepted. Hence, I suggest removing this inference on the Laccadive Basin and these sentences. Abstract and conclusions also may be modified accordingly.

10) Line 88: "…. Either side of the identified volcanic ridge". Please read this in view of my comment 9.

11) Lines 124-128: "*The trend of the intrusives and bathymetric highs in the study area follows the identified extensional trends…….., we noticed a series of volcanic mounds with a trend almost parallel to the CKE……… The observed trend correlates well with the crustal Bouguer anomaly map as well as the trap depth map*". The inference on trend of the intrusives in the Laccadive Basin derived only using two seismic section appears to be weak. Further, the bathymetric highs (consisting of seamounts, plateaus, knolls, hills, and guyots), most of which are interpreted to be associated with volcanism, are distributed randomly in different parts of the Laccadive Basin (please see Bijesh et al., 2018), but do not show any characteristic and systematic trend. Other than one trend representing CKE, any other such trends are not clearly visible from the crustal Bouguer anomaly map as well as the trap depth map provided in Figure 3. Therefore, this inference on the Laccadive Basin appears to be weak.

12) Line 152: "….. titled intrusive" or "tilted intrusive"? Please check.

13) Lines 17, 198, 225, and 227: The author's name "Bhattacharya, G." may be corrected as Bhattacharya, G.C."

14) Line 181: "GEBCO, C.G.". Please check, what is "C.G."?

---

## Author Response (AR1)

**List of all relevant changes made in the manuscript**

- The results section is rewritten to include only description of features
- The discussion section is re-arranged to give the model of opening of the Laccadive towards the end of the MS
- The introduction is re-written to more clearly describe the region picture and figure 1 is modified.
- All the corrections to figures suggested by the reviewers are incorporated.
- Three more seismic section is included in the supplementary material to show the continuity of the interpreted volcanic Ridge along the centre of the Laccadive Basin.

**Reviewer-1 Comments and Reply**

**Reply to major comments:**

9) Lines 79-83: It is mentioned that "*Further, a curvilinear trend of volcanic intrusive features is identified in the Centre of the Laccadive Basin parallel to the identified extensional trend. This trend is also observed in the gravity anomaly map as a broken chain of highs*". This inference does not appear to be convincing. First of all, you have only two seismic sections in which intrusives are mapped, therefore, with these two profiles, we can neither interpret the continuity of the features nor its arcuate trend. In addition, I am unable to identify any clear and convincing curvilinear trend from any of the maps presented in Figure 3". Your other interpretations on the ENE-WSW and NW-SE extension on the Laccadive Plateau are convincing as it is clearly observed from the seismic sections. Bringing the inference of Laccadive Basin trend actually dilute the quality of the paper. Therefore, this inference appears to be too weak to be accepted. Hence, I suggest removing this inference on the Laccadive Basin and these sentences. Abstract and conclusions also may be modified accordingly.

10) Line 88: "…. Either side of the identified volcanic ridge". Please read this in view of my comment 9.

Reply to comments 9 and 10.

We thank the reviewer for the comments. Comments 9 and 10 made by the reviewer critically analyses the trend and continuity of the volcanic intrusive feature identified towards the center of the Laccadive Basin. Two main arguments are put forward by the reviewer: 1) Only two seismic sections are showing intrusives which are not enough to interpret the continuity of the feature or the curvilinear trend. 2)The trend is not clearly visible in the gravity anomaly maps presented in figure 3. The arguments are very much valid and reasonable.

However, we inferred the presence of volcanic intrusive feature along the center of the basin by joint interpretation of the sediment deposition pattern, free-air gravity anomaly and seismic data. We noted that the sediment deposition pattern from Early Eocene to Early Miocene (Figure 4B) shows a divide along the center of the Basin and the intrusive pattern identified in the seismic sections falls along the same line. This led to the conclusion of the extent of the feature and its trend. Still we agree with the comment of the reviewer that this is not very convincing as the feature is only seen in the free-air anomaly map and not crustal Bouguer anomaly maps and its derivatives.

We have gone through some additional seismic lines in the region (sections given below) which very clearly show the intrusive features along the center of the Basin. The sediments are seen onlapping to the volcanic feature. This matches very well with the sediment deposition pattern from which we inferred the continuity of the feature. We infer from the new data that towards the center of the basin away from the shelf where the seismic data is located, the intrusive reached the sea bottom with subsequent sedimentation. This along with deeper water depth in the region mask the anomaly created by the intrusive in the crustal Bouguer anomaly map. This is argued as a reason for not observing the trend of the volcanic intrusive feature along the center of the basin in figure 3. We included these seismic sections in the supplementary material (Fig S3).

[Figure]

11) Lines 124-128: "*The trend of the intrusives and bathymetric highs in the study area follows the identified extensional trends……..., we noticed a series of volcanic mounds with a trend almost parallel to the CKE……… The observed trend correlates well with the crustal Bouguer anomaly map as well as the trap depth map*". The inference on trend of the intrusives in the Laccadive Basin derived only using two seismic section appears to be weak. Further, the bathymetric highs (consisting of seamounts, plateaus, knolls, hills, and guyots), most of which are interpreted to be associated with volcanism, are distributed randomly in different parts of the Laccadive Basin (please see Bijesh et al., 2018), but do not show any characteristic and systematic trend. Other than one trend representing CKE, any other such trends are not clearly visible from the crustal Bouguer anomaly map as well as the trap depth map provided in Figure 3. Therefore, this inference on the Laccadive Basin appears to be weak.

Reply to comment 11:
Three main arguments are put forwarded by the reviewer in this comment.
1) The inference on the trend of the intrusives in the Laccadive Basin derived only using two seismic section appears to be weak
2) Further, the bathymetric highs (consisting of seamounts, plateaus, knolls, hills, and guyots), most of which are interpreted to be associated with volcanism, are distributed randomly in different parts of the Laccadive Basin (please see Bijesh et al., 2018), but do not show any characteristic and systematic trend.
3) Other than one trend representing CKE, any other such trends are not clearly visible from the crustal Bouguer anomaly map as well as the trap depth map provided in Figure 3

The first point is discussed in the reply to comment 9 and 10 above.
Regarding the distribution of bathymetric highs, we believe that our idea was not clearly conveyed. We observed from the high-resolution bathymetry map that was published by Bijesh et al., 2018 that, most of the bathymetric highs south of Mangalore seems to be elongated in NE-SW direction. As mentioned in the comment, we also associate the high related to volcanism which we consider to be emplaced through weak zones or faults as discussed in section 5.2. Therefore, the orientation of the faults influenced the emplacement of the volcanic highs, as a result, the highs appear to be elongated roughly parallel to the trend of the Laccadive Basin. This strengthens the argument regarding the opening of Laccadive Basin.

Regarding the expression of bathymetric highs in the anomaly maps, we agree with the reviewer that the trends of the highs and lows (figure 3) are not prominent or continuous as that of CKE in the crustal Bouguer anomaly map and trap-depth map, however, the band-pass filtered crustal Bouguer anomaly and its first vertical derivative clearly show these features. Band-pass filtering is done to remove the deeper effects and enhance the crustal features. The trends are prominently seen in these two maps which indicate that these are shallow crustal level features (could be volcanic intrusives with few reaching the surface). The preferable elongation of these features indicates the direction of faults/weak zones through which the magma was able to migrate.

[Figure]

*Figure: This is high resolution bathymetry map presented by Bijesh et al., 2018 where the elongation of the features towards the south of Mangalore is clearly seen*

**Reply to minor comments:**

1) In the whole text: "Reunion" may be corrected as "Réunion" in throughout the manuscript.
Reply: We thank the reviewer for pointing this out. The text is corrected accordingly

3) Line 9: "pre-rift" may be modified as "pre-drift" since the age information cannot be derived from magnetic anomalies observed from rift stage crust, but possible only when these magnetic anomalies are formed by seafloor spreading (i.e., drifting"). Therefore, magnetic anomalies can provide only "pre-drift" juxtaposition.
Reply: We agree with the reviewer and the correction is made accordingly.

4) Lines 9-10: The detailed mapping of seafloor spreading magnetic anomalies in the conjugate Arabian and eastern Somali basins (spreading between India-Laxmi Ridge block and Seychelles) was published by Chaubey et al. (2002, Geological Society, London, Special Publication 195, pp. 71-85). The same may be quoted here. Although you mentioned geochronology here, the references are missing, please add the same.
Reply: We thank the reviewer for pointing out this and the references were added accordingly. More references of geochronology studies have been added as per the suggestion of the reviewer.

5) Figure Caption 1: "MR: Murray Ridge" may be deleted since you have not used this abbreviation in the figure, it is written in expanded form in the figure.
Reply: The correction is made as per the suggestion of the reviewer.

6) Lines 21-23: You mentioned *"...... whereas, more recent studies (Torsvik et al. (2013); Bhattacharya Yatheesh (2015) incorporate...."*. This sentence is misleading since Bhattacharya and Chaubey (2015) has not included Mauritius in their model. So, this sentence may be modified as "......whereas, more recent studies incorporate continental fragments like Laccadive Ridge (Bhattacharya and Yatheesh, 2015) or Mauritia, consisting of Mauritius, Southern Mascarene Plateau, Laccadive Plateau and Chagos Bank (Torsvik et al. (2013) between India and Madagascar in the India-Madagascar pre-drift scenario".
Reply: We agree with the reviewer for pointing out this and the sentence is modified according to the suggestion.

7) Line 37: "complicity" may be corrected as "complexity".
Reply: We thank the reviewer for pointing out this and the typo is corrected.

8) Line 39: The sentence "……. Laccadive Basin area will shed light on the margin's evolution …." May be modified as ""……. Laccadive Basin area will provide important constraints on the margin's evolution ….".
Reply: The sentence is modified according to the suggestion of the reviewer.

12) Line 152: "….. titled intrusive" or "tilted intrusive"? Please check.
Reply: We thank the reviewer for pointing out this and the typo is corrected.

13) Lines 17, 198, 225, and 227: The author's name "Bhattacharya, G." may be corrected as Bhattacharya, G.C."
Reply: We thank the reviewer for pointing out this and this is corrected in the Manuscript.

14) Line 181: "GEBCO, C.G.". Please check, what is "C.G."?
Reply: We thank the reviewer for pointing out this and the typo is corrected.

**Reviewer-2 Comments and Reply**

**Results section:**

**Comment:** Result item describes events instead of structures….i suggest that the authors separate description from interpretation when presenting results.

**Reply:** We agree with the reviewer and accordingly, the results section was rewritten as per the suggestion.

**Discussion part:**

**Comment:** The discussion part is not well organised…some information should be shifted to item 2 in order to help explaining the tectonics of the area…..giving tools to understand the results and interpretations further

**Comment:** 5.3 and 5.4 could be shifted to results or could be the start of the discussion item….leaving the plate tectonic picture to the end of the paper along with figure 5

**Reply:** The sections 5.3 and 5.4 are shifted after 5.1 and the section 5.2 describing the tectonic picture of the area is shifted towards the end of discussion as suggested by the reviewer. We thank the reviewer for the suggestion and we noticed that this change has improved the readability of the paper.

**Section 2 comments:**

**Comments:**

Contextualize the area and these separation events …. Expand chapter 2 and do some rewriting of the Introduction

The item 'Tectonics of the study area' describes the main features but actually not the tectonics

**Reply:** We thank the reviewer for the comment. The section is renamed as "Description of tectonic elements". A late Paleozoic fit of Gondwanaland showing the relative position of India and Madagascar is included in figure 1 to give a broader geodynamic view. Some rewriting of the introduction is done describing the major events that shaped the margin.

**Figures:**

**Comment:** The maps and figures in 3 & 4 are not extensively described and discussed

**Reply:** We have now included some more description regarding figure in the revised MS.

**Comment:** In figure 3D and 3E, the eastern LB and western LB are wrong

**Reply:** We thank the reviewer for pointing out this. The names were interchanged and now it is corrected in the fig 3.

**Comment:** In figure 4 please add absolute age interval – numbers in Ma for A and B

**Reply:** The figure is updated to add the absolute age interval in Ma for A, B and C.

**Comment:** Figure 5 is important since it summaries tectonic evolution in time slices…. Enlarge the maps and add legend …also add time interval

**Reply:** We thank the reviewer for pointing out this. The figure is modified as per the suggestion of the reviewer.

**Figure 1 comments:**

In figure 1 insert of Central Gondwana reconstructed, with the coastline of main continental blocks plus minor blocks in a Jurassic fit.

Add tectonic domains simplified (cratons and mobile belts) and suture/Shear zones since inheritance is described vaguely in the MS

In figure 1 – some features are lacking explanation – a legend would help

Reference to colours do not match map colours such as "Black solid lines are shear zones"... this is also true for figure 2

Add names Kochi and Mangalore to figure 1

I wonder if its possible to add an estimated COB or COT

Add the location of wells to one of the maps since they are cited throughout the text.

**Reply:** We thank the reviewer for the comments on figure 1. We have inserted a picture of Central Gondwana in figure 1. Further the names Kochi and Mangalore are now added to the MS and the locations some wells are also plotted in figure 1 and necessary corrections to figure 1 are made as per the suggestion.

**Scientific Question**

**Comment:** What would be a post-rift event? This would relate to which rift? Madagascar-India or Seychelles-India?

**Reply:** We thank the reviewer for the comment. By post-rift, wemeant the post-India Madagascar breakup event. Even though the margin was affected by two breakup events (the India-Madagascar and India-Seychelles), the later affected the northern part of the margin, north of Vengurla Arch. The region under consideration is towards the southern part and is related to India-Madagascar separation.

We now change post-rift to post-India-Madagascar separation throughout the MS.

**Comment:** Is there any evidence of a horizontal component for these graben-horst faults? Did the authors consider a transtensional component that would also accommodate the anti-clockwise rotation of India?

**Reply:** We thank the reviewer for the question. As such we did not notice any transtensional component in the present dataset. However, minor transtensional movement associated with rotation is not totally ruled out.

**Comment:** Can you add a map with the nature of the crust from this margin? Transitional, oceanic and continental stretched?

**Reply:** We thank the reviewer for the suggestion. However, as the present study does not involve any crustal modelling, we are not in a position to prepare such a map.

**Line by line comments**

**Comment:** Title – please add the age of the extension – "Cretaceous-Paleocene extension"

**Reply:** We thank the reviewer for the comment. The title is modified as per the suggestion of the reviewer.

**Comment:** Line 30 – add age of this hot spot

**Reply:** This is added in the revised MS

**Comment:** Line 31 – 20 m.y.r (instead of Ma)

**Reply:** This is added in the revised MS

**Comment:** Lines 31-33 – the 65 Ma, you mean the sediments above the volcanic trap?

**Reply:** The sentence is reframed for clarity.

**Comment:** Line 36 – what do you mean by post-Madagascar activity? Magmatic?

**Reply:** The sentence is reframed for clarity.

**Comment:** Line 37 – complicity?

**Reply:** We meant **complexity**, the typo is corrected in the MS.

**Comment:** Line 38 – What do you mean by the "state of the lithosphere".

**Reply:** We mean the Inheritance in the lithosphere. We have now corrected this in the MS.

**Comment:** Line 75 – "the prominent ENE-WSW extension observed", substitute by the NNW-SSE set of grabens observed, interpreted as a ENE-WSW extension…" and so goes through the results section.

**Reply:** We thank the reviewer for the comment. The results section is rewritten to describe the structures without any interpretation.

**Comment:** Lines 81 to 83 – call figure here.

**Reply:** The figure is reference in the revised MS.

**Comment:** Lines 87-88 – indicate the volcanic ridge on figure 4B, many features named on text are not well shown in the maps.

**Reply:** The figures are corrected accordingly.

**Comment:** Item 5.1 – Start please interpreting the data you present, before describing the bigger picture. My suggestion.

**Reply:** This part of the discussion is shifted towards the end and more description is added for clarity as per the suggestion of the reviewer.

**Comment:** Item 5.2 – I think there is some speculation in this part of the manuscript, unneeded. (A) "there are a large number of suture zones". Actually, there are not many, and the authors did not present a map of the terranes and sutures that are well known in the literature and might be related to the features of the margin, crustal scale reactivated structures.

**Reply:** The prominent shear zones or suture zones are shown in fig1 and fig5 (which shows major suture zones on both Indian and Madagascar side.). These features are not named in the text for brevity as we are not describing the features in detail. But as per the suggestion of the reviewer, the reference is now included in the text (Bhattacharya and Yatheesh 2015 and references therein).

**Comment:** Line 108 – Long (how long in km?)

**Reply:** We thank the reviewer for the question. The transform fault in question is from the reconstruction study by Shuhail et al 2018. We wanted here to emphasize the connection between the spreading in the Mascarene Basin and the area near ATTC, by highlighting that this has been suggested by earlier workers.

**Comment:** Line 111 – We believe? Please argument here.

**Reply:** We thank the reviewer for pointing out this. We have now elaborated the point with more arguments.

---

## Editor Decision (ED1)

**General comments:**

- Introduction assumes the reader knows the area well (see comments reviewer 2). In order to allow readers with limited knowledge on the area to understand what is being described, it would be needed to provide some context at the start, rather than directly describing the tectonic history of the area.

    o Names of tectonic features etc. need to be explained and shown on maps.

- The manuscript is short, which is nice overall, but often it is a bit too short and thus unclear (see comments)

- All figures are too small and need various adjustments(see specific comments)

- The supplement contains only 3 figures, and they are cited various times, which makes reading the manuscript a bit inconvenient. It may be better to simply add (the key parts of) these figures to the manuscript (perhaps even merged with the current manuscript figures, see also comments on figures).

- There seems to be some mixing of results and discussion. The results should present the raw observations, whereas interpretations and discussion should be provided in the discussion section.

- The order of the discussion chapter needs some adjustment.

- Timing of geological events is a big issue in this manuscript. There seems to be some circumstantial information that can help explain things, but there seems to be a lack of reliable data from the area itself (the only reliable data available seems to be the age of the sediments in the Laccadive Basin, but that does not allow us to interpret much beyond the development of the basin). As such, the interpretations and the proposed model seem rather speculative. It would be good to have additional information from wells and seismic interpretation (see comments below).

**Abstract**

- Line 1: "two-phase" may be better than "double"

- Line 3: perhaps use "with the Seychelles separating from India"

- Line 4: "is not discussed" seems to suggest that the topic is not discussed in this manuscript. How about "remains poorly constrained" or something similar?

- Line 6: it is not clear what the Mascarene Basin is, this should be fixed (see also comments of reviewer 2 regarding the general accessibility of the text to people who are not familiar with the region's geology).

- Line 7: Laccadive Ridge and Tellicherry Arch have the same issue as in Line 6

- Line 8: it should be "towards the south" I believe

- Line 8: "Plate reconstruction models" → these are your new models right? So use "Our new plate reconstruction models" or perhaps "our plate reconstruction modelling" to make this very clear. (now it reads like someone else did this work). Otherwise, use "previous plate reconstruction models" or so.

- Line 10: "Paleocene traps" or "a Paleocene trap"

- Line 11: "has been attributed" suggests this is someone else's idea, not something new in this paper. It should probably be "we attribute" or so

**Introduction**

- In general, the introduction starts with the geological history of the area, without introducing the general (present-day) setting. This is confusing, it would be much better to have a (quick) overview of the general features of the area, before diving into the tectonic history head-first. Also, some additional maps are needed to make this part work (see also comments on Fig. 1).

    o See also comments by reviewer 2 on the accessibility of the text for those who are not that familiar with the regional geology (of SW India).

- Line 16-17: here, the text should directly refer to Fig. 1 I would say (to illustrate the geology). In fact, it seems that Fig. 1 is not at all mentioned in the introduction? You should make sure to help the reader understand the geological context as much as possible, including ample references to figures.

- Line 18: consider using "this second break-up" for clarity

- Line 19: I believe it should be "fairly well established"?

- Line 23: it should probably be "the Southern Mascarene Plateau, the Laccadive Plateau, and the Chagos Bank"

- Line 23: the Chagos Bank is not indicated on any map it seems? Same for Mascarene Plateau?

- Line 25: "is" should be "are" I believe (or use "represent"?)

- Line 28: "wide-spread trap layers"

- Line 29-31: this sentence seems a bit out of place (it seems to describe the methods used in this manuscript). Can it be removed or rephrased a bit? → or include it in the last part of the introduction, where it would be good to quickly mention the methods used in this manuscript.

- Line 33: "long-time" → remove the hyphen?

- Line 33: "m.y.r." should probably be "Myr"

- Line 35-39: these sentences /motivation for this study seems a bit random to me. It is not that clear what is meant here, as these are rather different issues that are not clearly related

to each other (sediment ages vs. the overall complex geodynamic setting). It should be rephrased a bit. Some detailed comments:

- How would the absence of sediments fit with opening of the basin at 83 Ma (India-Madagascar break-up)? If anything, I would then expect that sediments are present, which is not the case?

  - And well CH-1-1 does in fact cross into older units? The sentence seems to suggest that the other wells were simply not deep enough to reach the relevant sedimentary layers?

- It seems to be strange to me that it would be a surprise to have older sediments below the Paleocene traps. Is it not to be expected that there would be older units/sediments below the traps?

- Line 36-39: this is a rather long sentence that seems to have some grammar issues, please double-check.

  - "new complexity" seems off (the complexity itself is not new, it's just that we don't/did not yet understand it I would say?) → "makes for a complex geodynamic setting" or so may work better here

  - "inheritance … before" seems off, how about "into the pre-existing lithospheric inheritance"?

- Line 36: "India-Madagascar separation"

- Line 39: use "the development of the Laccadive Basin" or something similar. The current wording seems to suggest there is a sedimentary formation called the "Laccadive Basin formation"

- Line 39-40: the same thing is stated in Line 42-44. I suggest removing it here to avoid duplication.

- Line 40: what kind of "evidence"? → see previous comment on mentioning the methods used in this manuscript. That way the reader can better appreciate where things are going.

- Line 42: "Understanding" seems a bit vague → what exactly needs to be understood?

- Line 43: "will provide" suggests this needs to be done in the future, but the start of the sentence seems to suggest it is already known. Please make very clear which of the two it is (e.g., use "future studying and time-stamping" or "event provides important constraints", respectively).

- Line 44: use "plate tectonic reconstruction studies" to make it clear what is reconstructed.

**Description of tectonic elements**

- Line 45: add "of the study area" to make it clear we are not talking about the region as a whole.

- o NB: several terms are used in this manuscript ("study area", "area under investigation", "area of interest"). I suggest choosing one and using it consistently throughout the manuscript.

- Again, make sure to refer to Fig. 1 early on

- See comments on Fig. 1 on the need for more maps

- Line 46-47: as it is written, it is not fully clear whether only the southern part of the Laccadive Basin is included → consider swapping the place of the ridge and the basin in this sentence. Also "in the offshore" seems incomplete?

- Line 49-50: the CKE is not indicated in Fig. 1 it seems? Please add all structures/locations mentioned in the text to relevant figures.

- Line 50-51: this is the first clear definition of the Laccadive Basin, 50 lines into the text. As this basin is in the title, it should be introduced very early on (in the first couple of lines).

  - o Ah, I now see that there is also a definition in line 25. Still, please consider the previous comments on "setting the stage" in the first sentences of the text.

- It seems that the CRS is not mentioned, even though it's a very important feature (for instance, it's the first topic of the discussion)? Please add some description here to prepare the reader.

**Data and Methods**

- Line 55-59: somehow the text is not that clear here: it is stated twice that seismic lines are used, apparently for the same purpose (?).

- Line 56: why not use the more recent 2023 GEBCO bathymetry data?

- Line 56: "the long-offset" → I believe that "the" should be deleted there. This goes for a number of places in the text, where "the" seems to indicate a very specific thing that is not really specified before in the text, and therefore seems a bit off. I hope this makes sense.

- Line 57: "provided" is a bit unclear, it seems to suggest that these data were simply taken from Unnikrishnan et al. 2023). These data cover the whole study area? It may be good to show the extent of the different datasets (in the supplement would be ok).

- Line 58: what are "intermediate" horizons? Please clarify in the text. (e.g. "and various horizons within the post-Paleocene sediments").

- Line 58: what is meant by "compiled"? Did you produce these sections yourself, or did you interpret them? Please rephrase to clarify.

- Line 62-63: a citation would be in order at the end of this sentence, or at the end of the previous one.

  - o Line 62: I suggest using "these two-way travel time (TWT) maps

- Note that TWT should be defined in line 61, as that is the first occurrence of the abbreviation.

- Line 65: add ", respectively" after "column".

- Line 68: only one seismic section, or multiple?

- Line 68: what is meant with "transferred"? you mean "identified on the gravity anomaly maps" I assume? Please rephrase.

- Line 84-85: the coast-parallel grabens are not shown? It may be better to just state that sedimentation is high along the coast.

**Results**

- Line 72-74: why are these extension directions interpreted as such? It seems that these en echelon graben arrangements may in fact indicate oblique kinematics, rather than orthogonal stretching. For example, the NNW-SSE oriented grabens could indicate ca. NNE-SSW extension. As such, you should be very careful with these statements here. In fact, this all goes into interpretation/discussion domain, and should be addresses in the discussion. The results are the place where the "clean" observations are presented.

- Line 77: how parallel to the extensional trend (or trends?) is this volcanic intrusive really? That is, what is the orientation of the extensional trend (not clearly defined)? Is it one "intrusive" or can we speak of a series of intrusive structures/bodies? Please rephrase where needed.

- Line 81-82: please annotate this channel in the figure, it's not that clear what is meant

- Line 83: the sedimentation is significant in the northern part of the Laccadive Basin, not overall. Please rephrase the text to better reflect this.

**Discussions**

- Line 88: I would use "Discussion"

- Line 89: see previous comment: what is the CRS? This needs to be clearly defined early on in the manuscript, as it seems to be very important

- Line 90-91: similar to the introduction, the reader is expected to remember everything about the local  (and regional) geology, and we directly dive into the geological history, rather than starting with the data and their implications to gradually build up to a regional picture. As a whole, section 5.1 seems out of place here → the discussion needs some reconstruction as to provide a logical story to present to the reader.

- Line 90-91: how do these data show that the development of the Laccavide Ridge occured after, and not during, India-Madagascar break-up?

- Line 91: what mainland is meant? India or Madagascar? Please indicate

- Line 91: how do we know it is passive extension? This needs to be explained

- Line 91-93: see previous comment on the interpretation of the extension directions as interpreted in this manuscript. Note also, that according to this interpretation, the southern part of the Laccadive basin would have seen yet another extension direction, given the orientation of the grabens. This is all too simplistic and needs more careful consideration.

    o Could it be that these basins are in fact of different age? See previous comment on the lack of interpreted horizons in the sections.

- Line 94: see comments on the use of/references to supplement data in the main text: this seems important data that should not be hidden in the supplement.

- Line 96-97: how do we know the age of the CRS?

- Line 97-100: how is the CRS defined? There are extensional structures further south, could these not simply be part of the CRS? Having some age constraints from seismic data could help here.

    o Regarding the different orientation of the grabens: an explanation could be that there was some inherited structural grain that got reactivated, forcing the development of these grabens in a different orientation than that what one would expect.

- Line 104: why suddenly use Mangalure and not the Tellicherry Arch as an indication here? (and why refer to Fig. 1, which is not relevant here?)

- Line 110-112: there is no beta-factor analysis provided? Please add this.

- Line 119-120: It is not clear what the median high is, and how it indicates opening of the basin after the Eocene (as the text seems to suggest now).

- Line 120: I would state "after the early Eocene" as it is not excluded that significant sedimentation (and thus basin development) initiated in the mid- or late Eocene.

- Line 122-123: would not the initial "patch" indicate the start of basin development?

- Line 126: what is meant with "by this time"? there is no clear or logical indication in the previous sentences to use this wording, please specify

- Line 126-128: it is not clear to me what information in this study justifies the correlation with the proposition of Unnikrishnan et al. (2018) that the Allepy Platform was formed during the Oligocene-Miocene. (what is meant by "formed"?) There is not seismic section provided that covers this platform, and I believe it is not even really addressed in the results? Please clarify in the text what is meant.

- Line 130: see previous comments on names of geological units/structures. Nowhere it is clear what the Mascarene Basin is.

- Line 130-145: the evolution proposed in this section seems nice, but also highly speculative as very little clear evidence is presented (either from the analysis in this paper, or from previous works). Various tectonic and geodynamic events are mentioned, which are not properly set up in the introduction. This all needs some work to make it more convincing.

Note also that most references are rather old, I assume there must be some newer works with the latest insights that could be used here.

- Line 130: in fact, it is not merely "near" but directly adjacent to the Mascarene Basin I believe? (the Mascarene Basin being the basin developing between India and Madagascar, if I understand it correctly)

- Line 134-140: see previous comments on the orientation of the grabens in the study area. It may be interesting to have a look at analogue and numerical modelling works that test the impact of inheritance during rifting. You can for instance have a look at the works by Henza et al., Molnar et al., Bonini et al., and Zwaan et al.

- Line 143-144: This seems a bit of a bold statement: what is the evidence for this? It should probably be toned down a bit.

- Line 145: is there any description of the age of the volcanics vs. the sediments in the basin? This would be an important observation from seismic sections to be included in the results (which it is not at the moment)

**Conclusion**

- Line 154: it should at least be specified what plume is meant here.

**Figure 1**

- This figure is much too small (especially the tectonic reconstruction), and the text is really not readable in large parts of both panels. Note also the varying font sizes → I strongly recommend standardizing font sizes.

- There is no ocean depth/topography scale it seems? Please add. (also in the supplement)

- It would be much better to include a general map (panel A from Fig. S1) to help the reader understand the various tectonic elements that are mentioned in the text, but not shown (e.g., Madagascar, Seychelles). Furthermore, It would be good to have a zoom-in map of the study area as well, to clearly show the tectonic elements described in section (2) of the text.

- Note that although the Laccadive Basin is in the title of the manuscript, there is no obvious indication of where it is situated. Instead, the left panel shows in large bold letters the Laccadive Ridge and Maldives.

- The left panel indicates the Laxmi Ridge (and SVP + DVP) as polygons, whereas elements such as the Laccadive Ridge and Maldives are not. This seems inconsistent. It would in fact be much better to show a simplified geological map (the general map from Fig. S1 could serve as a general introduction instead).

    o One thing that should probably be added: the Continent-Ocean transition, unless the Laccadive Basin is a (hyperextended) rift basin (this is not very clear)

- In the right panel, the area of interest is indicated with a red rectangle. This rectangle is however poorly visible (at least to me, I got slight red-green colorblindness). I would suggest using a black outline for the AOI, and using less thick greyish outlines for the continents.

    o Similarly the thick continental outlines drown out the break-up information.

- There are white and green lines used in the left panel. These are not very clearly distinguishable. Perhaps making the map larger would help, but also consider

- What is the definition of the Vengurla and Tellicherry Archs? I believe this is not really specified anywhere? Please clarify in the text.

**Figure 2**

- Also this figure is too small (including the text/annotation) and should be presented much larger. It may also be possible to rearrange the panels to allow for things to be made larger (i.e., move some of the sections below the map?)

- The color scale used in the map is a rainbow scale, which should be avoided (see the work by Fabio Crameri on the use of color in scientific publications). Moreover, the scale has no clear zero value color: a scale that has both positive and negative values should have a clear zero-value color to avoid artifacts and apparent structures.

- I see in this figure that there is an additional zoom-in to the study area. This becomes rather confusing, as there are now two zoom-ins (study areas?) of the general area shown in Fig. 1. It would be good to only use one extent to present the model results for consistency. It is now rather difficult to for instance compare the structures shown in Fig. 2 with those shown in Fig. 3 → are the lows in fact tracing the interpreted grabens? It

    o Also, there should be an indication in the caption that the location of this map is shown in Fig. 1.

- The sections miss an indication that the seconds are in TWT. At the least, this should be indicated in the caption (including a definition of TWT).

- Using circles to indicate circles is a bit confusing → one may mistake it for a zoom-in. It would probably better to just use an arrow, or perhaps a dotted circle instead.

- The white arrow indicating the Tellicherry Arch is poorly visible. Consider using another color. (same for other figures)

- Caption: what does "CRS represents the Cannanore Rift System as identified by DGH" mean? What is "DGH" an abbreviation of? Please specify.

- Note that the "broken brown line" is very poorly visible in the map. Please improve this.

    o Note that the line is in fact not broken (?)

- The horizontal scale of BB' and CC' is different from those in the other sections (which appear to also represent variable lengths in map view. It may look esthetically pleasing to

have these sections in the figure all at the same size, but it does not properly represent the natural situation and the relations between these sections. Please rescale things.

- o This is also relevant to Fig. S2

- Seismic line labeling: why are some of these lines labels with numbers, and other with letters? Please standardize things.

- Overall, only faults are interpreted in these seismic sections. Is there no data whatsoever about ages etc.? There is a mention of various boreholes in the area, so I would think this could be added? → like is done for sections 1-3 in the supplement.

- o Note that there is various annotation in sections 1-3 that is not explained anywhere (no legend)

- Wy are the grabens in the Trivandrum Terrace area indicated in white? They are barely visible. Please use the same color as used to the west.

- o Same for the NW corner of the map

- Upon closer inspection of the seismic sections: it seems that there are many faults that were not interpreted. Why not? In fact, I realized that the Laccadive Basin is the study area, but there is not one section that clearly shows the general characteristics of the basin (it is a rifted basin right?) → I would suggest having a look at (the figures of) Gireesh & Pandey (2014) → Open Access link: https://www.researchgate.net/publication/260213497

**Figure 3**

- Panel (E) is described as a tectonic map in the caption, which it is not really? (panel F seems to be?)

- o Note that panel F is not a map of beta-values, as described in the caption

- o Overall, panels (E) and (F) seem to represent general interpretations, rather than results, and should as such be made into separate discussion figures.

- See comments on the use of scientific color (scales) in Fig. 2. These are also relevant here; the color scales in Fig. 3 seem inappropriate.

- o Color scale units are not always aligned in the same way (compare panel A with the other panels.

- The lows are indicated using red lines. These lines are poorly visible: please use another indication (e.g., black dotted lines).

- o The same for the CKE in green.

- Caption: the abbreviations of TA and TT are nor provided, please add these

- Caption: the repeated "with all identifications" is a bit vague. Consider using "with all identified/interpreted structures"

- In panels B-F, but not in A, there are additional lines in the SE. What do these represent (it is not clear what "shelfal tectonic elements" are, and why ther are not indicated in panel A)?

**Figure 4**

- Somehow the study area has a different extent than that in Fig. 3? Please standardize the study area extent in your maps for consistency.

- See previous comments on the use of colors. This needs to be improved here.

  o It would be best to use the same scale for panels A-C, to allow for easy comparison between the different time intervals

- I suggest using a broken line or something less dominant to indicate the sediment patch in panels A and B.

**Figure 5**

- This figure needs to be larger to better show the details

- Stage III covers no less than 40 Myr, but seems to show a snapshot of the initial Laccadive Basin opening (around 60 Myr?). I strongly suggest avoiding having such time ranges in these panels, as it is confusing.

- Stage IV: I would simply remove Madagascar to avoid confusion. The way it is now shown, it seems to suggest India and Madagascar are pretty close to each other, with the black line representing a mid-oceanic ridge.

- It would be useful to add some annotation highlighting the important events in the system.

  o Note the timing of the various events: how do we know the age of the rifting that is attributed to stage II? This is not really specified/justified in the text?

  o See also the comment on the last part of the discussion: it would be good to

- I suggest moving the text "Stage-I" etc. in each panel to the bottom-right corner (it seems poorly aligned at the moment.

  o Also, the header of the Stage-I panel seems not properly aligned

- Caption: please provide the meaning of ATTC and CKE (each abbreviation in a caption/figure needs to be explained in the caption [of that figure]).

---

## Author Response (AR2)

**General Comments:**

- Introduction assumes the reader knows the area well (see comments reviewer 2). In order to allow readers with limited knowledge on the area to understand what is being described, it would be needed to provide some context at the start, rather than directly describing the tectonic history of the area.
- Names of tectonic features etc. need to be explained and shown on maps.

- The manuscript is short, which is nice overall, but often it is a bit too short and thus unclear (see comments)

- All figures are too small and need various adjustments (see specific comments)

- The supplement contains only 3 figures, and they are cited various times, which makes reading the manuscript a bit inconvenient. It may be better to simply add (the key parts of) these figures to the manuscript (perhaps even merged with the current manuscript figures, see also comments on figures).

- There seems to be some mixing of results and discussion. The results should present the raw observations, whereas interpretations and discussion should be provided in the discussion section.

- The order of the discussion chapter needs some adjustment.

- Timing of geological events is a big issue in this manuscript. There seems to be some circumstantial information that can help explain things, but there seems to be a lack of reliable data from the area itself (the only reliable data available seems to be the age of the sediments in the Laccadive Basin, but that does not allow us to interpret much beyond the development of the basin). As such, the interpretations and the proposed model seem rather speculative. It would be good to have additional information from wells and seismic interpretation (see comments below).

Reply:

We thank the editor for going through the manuscript in detail and providing many useful suggestions and comments. We have incorporated all the suggestions made by the editor, which greatly improved the readability and quality of the MS. The suggestions regarding the use of scientific color scale and the references to numerical/analog modelling studies were greatly useful.

We have added more details and figures to the introduction section describing the present-day setting and defining important features like the Laccadive Ridge, Laccadive Basin, the Mascarene Basin, Tellicherry Arch etc. The manuscript (MS) is elaborated by adding more material to the introduction and discussion sections. In addition to this, the discussion section is rearranged and elaborated. Similarly, all the figures are enlarged and more figures are added to the MS and supplementary section. We have tried to explain the evolution of the basin as clearly as possible. The detailed reply to comments line by line is given below:

**Abstract**

**Comment:** Line 1: "two-phase" may be better than "double"
Reply: We thank the editor for the comment. The suggestion is included.

**Comment:** Line 3: perhaps use "with the Seychelles separating from India"
Reply: We incorporated this suggestion.

**Comment:** Line 4: "is not discussed" seems to suggest that the topic is not discussed in this manuscript. How about "remains poorly constrained" or something similar?
Reply: We thank the editor for the comment. We now changed the sentence.

**Comment:** Line 6: it is not clear what the Mascarene Basin is, this should be fixed (see also Comments of editor 2 regarding the general accessibility of the text to people who are not familiar with the region's geology).
**Comment:** Line 7: Laccadive Ridge and Tellicherry Arch have the same issue as in Line 6
Reply: The description about the present-day setting and regional geology (especially Mascarene Basin, Laccadive Ridge, Tellicherry Arch) is now included in the revised MS.

**Comment:** Line 8: it should be "towards the south" I believe
Reply: We agree. We made necessary change.

**Comment:** Line 8: "Plate reconstruction models" → these are your new models right? So use "Our new plate reconstruction models" or perhaps "our plate reconstruction modelling" to make this very clear. (now it reads like someone else did this work). Otherwise, use "previous plate reconstruction models" or so.
Reply: We meant to say previous plate reconstruction models. The MS is revised accordingly.

**Comment:** Line 10: "Paleocene traps" or "a Paleocene trap"
Reply: It is Paleocene trap. The MS is revised accordingly.

**Comment:** Line 11: "has been attributed" suggests this is someone else's idea, not something new in this paper. It should probably be "we attribute" or so
Reply: The sentence is rephrased for clarity.

**Introduction**

**Comment:** In general, the introduction starts with the geological history of the area, without introducing the general (present-day) setting. This is confusing, it would be much better to have a (quick) overview of the general features of the area, before diving into the tectonic history head-first. Also, some additional maps are needed to make this part work (see also Comments on Fig. 1).
- See also comments by reviewer 2 on the accessibility of the text for those who are not that familiar with the regional geology (of SW India).

**Comment:** Line 16-17: here, the text should directly refer to Fig. 1 I would say (to illustrate the geology). In fact, it seems that Fig. 1 is not at all mentioned in the introduction? You should make sure to help the reader understand the geological context as much as possible, including ample references to figures.

Reply: We thank the editor for the above comments. A brief description about the present-day setting of the northwestern Indian ocean is now included in the MS. Moreover, two figures, one showing the general setting of the northwestern Indian Ocean and Gondwanaland in late Paleozoic fit and the other showing detailed tectonic map of the western continental margin of India and the Mascarene Basin is now added to the introduction. Care has been taken to introduce and define Laccadive Ridge, Laccadive Basin and Tellicherry Arch in the introduction itself.

**Comment:** Line 18: consider using "this second break-up" for clarity
Reply: The MS is revised accordingly.

**Comment:** Line 19: I believe it should be "fairly well established"?
Reply: Agree. The MS is revised accordingly.

**Comment:** Line 23: it should probably be "the Southern Mascarene Plateau, the Laccadive Plateau, and the Chagos Bank"
Reply: We agree. The MS is revised accordingly.

**Comment:** Line 23: the Chagos Bank is not indicated on any map it seems? Same for Mascarene Plateau?
Reply: Figure 1A now shows Chagos bank (CB). Saya-del-Malha and Nazareth Bank together is the Mascarene plateau (fig. 1A & 2B).

**Comment:** Line 25: "is" should be "are" I believe (or use "represent"?)
Reply: The MS is modified to correct this.

**Comment:** Line 28: "wide-spread trap layers"
Reply: The MS is revised accordingly.

**Comment:** Line 29-31: this sentence seems a bit out of place (it seems to describe the methods used in this manuscript). Can it be removed or rephrased a bit? → or include it in the last part of the introduction, where it would be good to quickly mention the methods used in this manuscript.
Reply: We agree with the editor. We have removed this part for clarity.

**Comment:** Line 33: "long-time" → remove the hyphen?
Reply: The MS is revised accordingly.

**Comment:** Line 33: "m.y.r." should probably be "Myr"
Reply: All through the MS, we prefer to give as Ma. The MS is revised accordingly.

**Comment:** Line 35-39: these sentences /motivation for this study seems a bit random to me. It is not that clear what is meant here, as these are rather different issues that are not clearly related to each other (sediment ages vs. the overall complex geodynamic setting). It should be rephrased a bit. Some detailed Comments:

- How would the absence of sediments fit with opening of the basin at 83 Ma (India-Madagascar break-up)? If anything, I would then expect that sediments are present, which is not the case?
- And well CH-1-1 does in fact cross into older units? The sentence seems to suggest that the other wells were simply not deep enough to reach the relevant sedimentary layers?
- It seems to be strange to me that it would be a surprise to have older sediments below the Paleocene traps. Is it not to be expected that there would be older units/sediments below the traps?

Reply: we thank the editor for the questions. With regard to the questions above, there are some details to be considered, firstly C-H-1-1 well lies on the continental shelf close to the coast at shallow water depth (250 m Approx.). Secondly in the **Laccadive Basin** Mesozoic sediments are not encountered until now. But the plate-tectonic reconstruction studies based on magnetic anomaly identifications and volcanism show that separation of India and Madagascar started around 83 Ma. It is a possibility that there can be presence of Mesozoic sediments below the trap layer. Another alternative scenario is that the Laccadive Basin opened up later as implied by data in this study, in which case the probability of finding older sediments maybe limited to pockets along the shelf which opened up during the India-Madagascar separation.

Line 36-39: this is a rather long sentence that seems to have some grammar issues, please double-check.
- "new complexity" seems off (the complexity itself is not new, it's just that we don't/did not yet understand it I would say?) → "makes for a complex geodynamic setting" or so may work better here
- "inheritance … before" seems off, how about "into the pre-existing lithospheric inheritance"?

Reply: We thank the editor for the comment. The changes are made accordingly.

**Comment:** Line 36: "India-Madagascar separation"
Reply: The changes are made accordingly.

**Comment:** Line 39: use "the development of the Laccadive Basin" or something similar. The current wording seems to suggest there is a sedimentary formation called the "Laccadive Basin formation"
Reply: The changes are made accordingly.

**Comment:** Line 39-40: the same thing is stated in Line 42-44. I suggest removing it here to avoid duplication.
Reply: The changes are made as per the suggestion of the editor.

**Comment:** Line 40: what kind of "evidence"? → see previous comment on mentioning the methods used in this manuscript. That way the reader can better appreciate where things are going.
Reply: The sentence is rephrased for clarity and better understanding.

**Comment:** Line 42: "Understanding" seems a bit vague → what exactly needs to be understood?
**Comment:** Line 43: "will provide" suggests this needs to be done in the future, but the start of the sentence seems to suggest it is already known. Please make very clear which of the two it is

(e.g., use "future studying and time-stamping" or "event provides important constraints", respectively).

**Comment:** Line 44: use "plate tectonic reconstruction studies" to make it clear what is reconstructed.

Reply: We thank the editor for pointing out these. The sentence is reframed as per the suggestion of the editor.

**Description of tectonic elements**

**Comment:** Line 45: add "of the study area" to make it clear we are not talking about the region as a whole.

- NB: several terms are used in this manuscript ("study area", "area under investigation", "area of interest"). I suggest choosing one and using it consistently throughout the manuscript.

Reply: We thank the editor for pointing out this. The changes are made as per the suggestion of the editor. We have now used study area throughout the MS.

**Comment:** Again, make sure to refer to Fig. 1 early on

**Comment:** See comments on Fig. 1 on the need for more maps

Reply: We included few additional in the introduction to better explain the present-day setting and tectonic history.

**Comment:** Line 46-47: as it is written, it is not fully clear whether only the southern part of the Laccadive Basin is included → consider swapping the place of the ridge and the basin in this sentence. Also "in the offshore" seems incomplete?

Reply: The Laccadive ridge extends further north upto around $16^0$ N. We mean to say that the study area (Laccadive Basin) lies between the southern part of the Ridge (south of Tellicherry Arch) and the continental shelf of India. The part in the offshore is redundant and we have deleted it. Hence it is mentioned *"southern part of the Laccadive ridge and the Laccadive Basin"*.

**Comment:** Line 49-50: the CKE is not indicated in Fig. 1 it seems? Please add all structures/locations mentioned in the text to relevant figures.

Reply: We thank the editor for pointing out this. The figures are now modified and all the relevant features mentioned in the MS are now included.

**Comment:** Line 50-51: this is the first clear definition of the Laccadive Basin, 50 lines into the text. As this basin is in the title, it should be introduced very early on (in the first couple of lines).

- Ah, I now see that there is also a definition in line 25. Still, please consider the previous comments on "setting the stage" in the first sentences of the text.

Reply: We have modified the introduction to explain the present-day setting.

**Comment:** It seems that the CRS is not mentioned, even though it's a very important feature (for instance, it's the first topic of the discussion)? Please add some description here to prepare the reader.

Reply: We thank the editor for pointing out this. The CRS is identified by Director General of Hydrocarbons, a body under the government of India. They used 2D and 3D seismic data to identify the features but the details of the seismic sections are not available in the public domain.

The identified extensional features north of Tellicherry Arch falls within CRS and the identification by DGH stops abruptly with a sharp cut where a different extensional event was identified in this study. Hence, we believe that the extensional features north of Tellicherry Arch forms the CRS and towards the south of it represent a different tectonic event. The link to DGH report is given in the MS and is accessible to the public.

**Data and Methods**

**Comment:** Line 55-59: somehow the text is not that clear here: it is stated twice that seismic lines are used, apparently for the same purpose (?).
Reply: The first mention of the seismic lines refers to the section which are presented in this paper. The second mention is the vast amount of seismic data which is used to derive the sediment thickness map of the basin. This is industry data, the details of the lines used to create the sediment thickness map are given in figure S3. (after Unnikrishnan et al., 2023)

**Comment:** Line 56: why not use the more recent 2023 GEBCO bathymetry data?
Reply: We thank the editor for pointing out this. Major portion of the work was done last year and hence the old dataset. We compared the two datasets in the study area and they seem to be matching well except some isolated areas but it can be seen that this does not affect the conclusions from the study.

[Figure]

**Figure:** Comparison between GEBCO2020 and GEBCO2023 version of bathymetry data

**Comment:** Line 56: "the long-offset" → I believe that "the" should be deleted there. This goes for a number of places in the text, where "the" seems to indicate a very specific thing that is not really specified before in the text, and therefore seems a bit off. I hope this makes sense.
Reply: We thank the editor for the comments. We have made changes accordingly.

**Comment:** Line 57: "provided" is a bit unclear, it seems to suggest that these data were simply taken from Unnikrishnan et al. 2023). These data cover the whole study area? It may be good to show the extent of the different datasets (in the supplement would be ok).

Reply: Yes. The data refers is the vast amount of data which is used to derive the sediment thickness map of the basin. This is industry data, the details of the lines used to create the sediment thickness map is given in Unnikrishnan et al., 2023. The diagram showing the areal extend of data and the seismic lines used for the preparation of the TWT maps is now included in the supplementary file S3.

**Figure:** The seismic profiles used to prepare the Two-way time (TWT) maps. (Reproduced from Unnikrishnan et al., 2023)

**Comment:** Line 58: what are "intermediate" horizons? Please clarify in the text. (e.g. "and various horizons within the post-Paleocene sediments").
Reply: The intermediate horizons refer to the early Paleocene, early Eocene, early Miocene respectively. This is mentioned in the second paragraph of the section data and methods.

**Comment:** Line 58: what is meant by "compiled"? Did you produce these sections yourself, or did you interpret them? Please rephrase to clarify.
Reply: We thank the editor for asking this. We gathered interpreted seismic sections in the study area to support our observations and synthesis. We have not interpreted the data but have taken the interpretations given in those papers. The interpretations given by these earlier studies support our model presented.

**Comment:** Line 62-63: a citation would be in order at the end of this sentence, or at the end of the previous one.
- Line 62: I suggest using "these two-way travel time (TWT) maps
- Note that TWT should be defined in line 61, as that is the first occurrence of the abbreviation.
Reply: We thank the editor for pointing out this. The changes are made as per the suggestion of the editor.

**Comment:** Line 65: add ", respectively" after "column".
Reply: The changes are made as per the suggestion of the editor.

**Comment:** Line 68: only one seismic section, or multiple?
Reply: The changes are made as per the suggestion of the editor.

**Comment:** Line 68: what is meant with "transferred"? you mean "identified on the gravity anomaly maps" I assume? Please rephrase.
Reply: We thank the editor for pointing out this. By 'transfered' we mean that, the location of the features identified were transferred to the gravity anomaly maps and their continuity were mapped.

**Comment:** Line 84-85: the coast-parallel grabens are not shown? It may be better to just state that sedimentation is high along the coast.
Reply: We thank the editor for pointing out this. The changes are made as per the suggestion of the editor.

**Results**

**Comment:** Line 72-74: why are these extension directions interpreted as such? It seems that these en echelon graben arrangements may in fact indicate oblique kinematics, rather than orthogonal stretching. For example, the NNW-SSE oriented grabens could indicate ca. NNE-SSW extension. As such, you should be very careful with these statements here. In fact, this all goes into interpretation/discussion domain, and should be addresses in the discussion. The results are the place where the "clean" observations are presented.
Reply: We thank the editor for pointing out this and the suggestion of references to analog and numerical modelling studies were greatly helpful. We agree to the comment that the first rifting event greatly influences the grabens developed during a later event. We have shifted the interpretation part to the next section incorporating the suggestions by the editor.

**Comment:** Line 77: how parallel to the extensional trend (or trends?) is this volcanic intrusive really? That is, what is the orientation of the extensional trend (not clearly defined)? Is it one "intrusive" or can we speak of a series of intrusive structures/bodies? Please rephrase where needed.
Reply: It is not a single intrusive but a series of intrusives and the extensional trend is the NNE-SSW trend identified south of Tellicherry Arch. The sentence is reframed for clarity.

**Comment:** Line 81-82: please annotate this channel in the figure, it's not that clear what is meant
Reply: The channel is now annotated in the figure. We mean to say that the sediment channel may represent the very initial stage of opening of the Laccadive Basin.

**Comment:** Line 83: the sedimentation is significant in the northern part of the Laccadive Basin, not overall. Please rephrase the text to better reflect this.
Reply: The changes are made as per the suggestion of the editor.

**Discussion**

**Comment:** Line 88: I would use "Discussion"
Reply: This is corrected in the MS.

**Comment:** Line 89: see previous comment what is the CRS? This needs to be clearly defined early on in the manuscript, as it seems to be very important

Reply: As mentioned earlier, this is now clarified in the MS in the discussion part (see also reply to earlier comment).

**Comment:** Line 90-91: similar to the introduction, the reader is expected to remember everything about the local (and regional) geology, and we directly dive into the geological history, rather than starting with the data and their implications to gradually build up to a regional picture. As a whole, section 5.1 seems out of place here→ the discussion needs some reconstruction as to provide a logical story to present to the reader.

Reply: We thank the editor for pointing out this. After going through the MS in the light of the comments, we understood this drawback. Now we have improved the discussion section first by start interpreting the results then going to regional picture. Some sections are merged with the first part of discussion and some rearrangement is also carried out (Please refer to paragraph one of discussion)

**Comment:** Line 90-91: how do these data show that the development of the Laccadive Ridge occurred after, and not during, India-Madagascar break-up?

**Comment:** Line 91: what mainland is meant? India or Madagascar? Please indicate

**Comment:** Line 91: how do we know it is passive extension? This needs to be explained

Reply: We thank the editor for the above questions and comment. We agree that the data presented do not indicate the development of the Laccadive Ridge after India-Madagascar breakup. It is highly probable that this happened during India-Madagascar separation as the study area in an extensional regime during that time (also evidenced by the presence of Mesozoic sediments in CH-1-1 well). Similarly, there is no data to state it is a passive extension. So, we have reframed the sentence to correct these mistakes and bring more clarity.

**Comment:** Line 91-93: see previous comment on the interpretation of the extension directions as interpreted in this manuscript. Note also, that according to this interpretation, the southern part of the Laccadive basin would have seen yet another extension direction, given the orientation of the grabens. This is all too simplistic and needs more careful consideration.

• Could it be that these basins are in fact of different age? See previous comment on the lack of interpreted horizons in the sections.

Reply: We thank the editor for the above comment. We have not carried out interpretation of seismic horizons in the study hence we cannot comment on that. The discussion on this matter is now included in the MS.

**Comment:** Line 94: see comments on the use of/references to supplement data in the main text: this seems important data that should not be hidden in the supplement.

Reply: As mentioned earlier more figures are now included in the MS and some additional figures in the supplementary material.

**Comment:** Line 96-97: how do we know the age of the CRS?

Reply: We do not know the age of CRS. From the orientation of the extensional features on CRS and the spreading direction recorded in the magnetic anomalies in the Mascarene Basin it can be safely seen that they can be correlated. Moreover, this is also parallel to Dharwar trend which is a

prominent structural trend onshore. So, it is considered in this study that CRS formed during India-Madagascar separation.

**Comment:** Line 97-100: how is the CRS defined? There are extensional structures further south, could these not simply be part of the CRS? Having some age constraints from seismic data could help here.

- Regarding the different orientation of the grabens: an explanation could be that there was some inherited structural grain that got reactivated, forcing the development of these grabens in a different orientation than that what one would expect.

Reply: Here we consider the change in orientation to represent a different extensional event. This conclusion is made because of two major reasons. 1) the entire area has a common evolutionary history but the trend north and south of Tellicherry Arch is markedly different. 2)There is no prominent NNE-SSW trend in the onshore region to explain the extensional features south of Tellicherry Arch whereas the extensional features in the north of the Arch follows the dominant Dharwar trend. More discussion about these aspects are now included in the discussion part

**Comment:** Line 104: why suddenly use Mangalure and not the Tellicherry Arch as an indication here? (and why refer to Fig. 1, which is not relevant here?)
Reply: We thank the editor for pointing out this. This is now corrected in the MS. (Please refer to figure S2 in the revised MS)

**Comment:** Line 110-112: there is no beta-factor analysis provided? Please add this.
Reply: As this is now included in the MS.

**Comment:** Line 119-120: It is not clear what the median high is, and how it indicates opening of the basin after the Eocene (as the text seems to suggest now).
Reply: The median high refers to the chain of volcanic intrusive identified along the centre of the basin. It is seen in the sediment deposition map (from Early Eocene to Early Miocene) that the sediments are deposited on either side of this feature. This indicate that the feature was present during this time (intrusives already got emplaced forming a high). The high lies along the centre of the Basin and deposition is seen on either side of the feature. Hence, we infer that the basin opened after Early Eocene.

**Comment:** Line 120: I would state "after the early Eocene" as it is not excluded that significant sedimentation (and thus basin development) initiated in the mid- or late Eocene.
Reply: We agree with the editor and changes are made in the MS.

**Comment:** Line 122-123: would not the initial "patch" indicate the start of basin development?
Reply: Yes. The initial patch can indicate the start of the basin development. We thank the editor for this comment. This is now added in the discussion part.

**Comment:** Line 126: what is meant with "by this time"? there is no clear or logical indication in the previous sentences to use this wording, please specify
Reply: The sentence was unnecessary and is removed.

**Comment:** Line 126-128: it is not clear to me what information in this study justifies the correlation with the proposition of Unnikrishnan et al. (2018) that the Alleppy Platform was

formed during the Oligocene-Miocene. (what is meant by "formed"?) There is not seismic section provided that covers this platform, and I believe it is not even really addressed in the results? Please clarify in the text what is meant.

Reply: Unnikrishnan et al. (2018) identified the Alleppey platform as a continental fragment and inferred its development during the Oligocene-Miocene period. Alleppey platform is located adjacent to the Laccadive Basin and hence the development of the basin and the platform is related. The timing of the development of the Alleppey platform given by Unnikrishnan et al. (2018) closely agrees with the inferred timing of the opening of the Laccadive Basin from this study. This is now clarified in the MS.

**Comment:** Line 130: see previous comments on names of geological units/structures. Nowhere it is clear what the Mascarene Basin is.

Reply: As mentioned earlier this is now included in the MS (figure 1A and 2B).

**Comment:** Line 130-145: the evolution proposed in this section seems nice, but also highly speculative as very little clear evidence is presented (either from the analysis in this paper, or from previous works). Various tectonic and geodynamic events are mentioned, which are not properly set up in the introduction. This all needs some work to make it more convincing. Note also that most references are rather old, I assume there must be some newer works with the latest insights that could be used here.

Reply: The section is expanded adding more details to it for clarity. We have included all the relevant recent studies in the MS.

**Comment:** Line 130: in fact, it is not merely "near" but directly adjacent to the Mascarene Basin I believe? (the Mascarene Basin being the basin developing between India and Madagascar, if I understand it correctly)

Reply: We agree with the editor. This is now corrected in the MS.

**Comment:** Line 134-140: see previous comments on the orientation of the grabens in the study area. It may be interesting to have a look at analogue and numerical modelling works that test the impact of inheritance during rifting. You can for instance have a look at the works by Henza et al., Molnar et al., Bonini et al., and Zwaan et al.

Reply: We thank the editor for this suggestion and the references. We have incorporated inputs from the studies in the paper. This was very helpful and added to the paper.

**Comment:** Line 143-144: This seems a bit of a bold statement: what is the evidence for this? It should probably be toned down a bit.

Reply: The sentence is reframed as per the suggestion.

**Comment:** Line 145: is there any description of the age of the volcanics vs. the sediments in the basin? This would be an important observation from seismic sections to be included in the results (which it is not at the moment)

Reply: Unfortunately, there are no ages of volcanics available from the study area. In seismic section presented in fig. 4 it can be seen as sediments onlapping on the intrusive features. This indicates that the sediments are younger compared to the volcanic intrusion.

**Conclusion**

**Comment:** Line 154: it should at least be specified what plume is meant here.

Reply: The change is made as per the suggestion of the editor.

**Figure 1**

**Comment:** This figure is much too small (especially the tectonic reconstruction), and the text is really not readable in large parts of both panels. Note also the varying font sizes → I strongly recommend standardizing font sizes.

**Comment:** It would be much better to include a general map (panel A from Fig. S1) to help the reader understand the various tectonic elements that are mentioned in the text, but not shown (e.g., Madagascar, Seychelles). Furthermore, it would be good to have a zoom-in map of the study area as well, to clearly show the tectonic elements described in section (2) of the text.

**Comment:** Note that although the Laccadive Basin is in the title of the manuscript, there is no obvious indication of where it is situated. Instead, the left panel shows in large bold letters the Laccadive Ridge and Maldives.

Reply: We thank the editor for pointing out these. As replied earlier, additional figures are given in the introduction (refer to figure 1 and figure 2). Figure 1 now shows the configuration of northwestern Indian Ocean and the location of the study area in Gondwanaland in late Paleozoic fit. This gives a regional picture. Figure 2 shows the study area (that is the Laccadive Ridge, the Laccadive Basin and adjoining areas) and the Mascarene Basin in detail. In addition to this, the figures are made bigger. Laccadive Basin is now very clearly shown in figure 2B.

**Comment:** There is no ocean depth/topography scale it seems? Please add. (also, in the supplement)

Reply: This is now added in the figures.

**Comment:** The left panel indicates the Laxmi Ridge (and SVP + DVP) as polygons, whereas elements such as the Laccadive Ridge and Maldives are not. This seems inconsistent. It would in fact be much better to show a simplified geological map (the general map from Fig. S1 could serve as a general introduction instead). One thing that should probably be added: the Continent-Ocean transition, unless the Laccadive Basin is a (hyperextended) rift basin (this is not very clear)

Reply: We thank the editor for pointing out this. Earlier the Laxmi Ridge and SVP was given as polygons since they are not very clearly visible in the bathymetry map and the boundaries are identified from free-air anomaly and seismic studies respectively and the boundary of DVP from geological mapping. Whereas the Laccadive Ridge and the Maldives ridge is identified by bathymetry. The figure from S1 is now added to the MS in figure 1. A detailed tectonic map of the study area is given in figure 2A. The continent-ocean boundary (COB) is marked towards the west of Laccadive Ridge as COB since most of the studies indicate that the LR is underlain by hyperextended continental crust (Murty et al., 1999; Gireesh and Pandey 2014; Unnikrishnan et al., 2023).

**Comment:** In the right panel, the area of interest is indicated with a red rectangle. This rectangle is however poorly visible (at least to me, I got slight red-green colorblindness). I would suggest using a black outline for the AOI, and using less thick greyish outlines for the continents. Similarly, the thick continental outlines drown out the break-up information.

Reply: We thank the editor for pointing out this. The figure is enlarged and modified accordingly.

**Comment:** There are white and green lines used in the left panel. These are not very clearly distinguishable. Perhaps making the map larger would help, but also consider
Reply: We thank the editor for pointing out this. The map is enlarged for clarity and the limits are decreased to better show the features discussed in the section "Tectonics of the study Area".

**Comment:** What is the definition of the Vengurla and Tellicherry Archs? I believe this is not really specified anywhere? Please clarify in the text.
Reply: Using the huge volume of industry seismic data, ONGC has identified several Basement Arches orthogonal to the coast and the WCMI into several offshore sub-basins. Vengurla Arch and Tellicherry Arch are basement highs identified in earlier studies which is suggested to have implications on basin segmentation (Biswas 1989).

**Figure 2**

**Comment:** Also, this figure is too small (including the text/annotation) and should be presented much larger. It may also be possible to rearrange the panels to allow for things to be made larger (i.e., move some of the sections below the map?)
Reply: We thank the editor for the suggestion. The figure (figure 4) is enlarged and a crustal Bouguer anomaly map of the area is added for showing the continuity of features north and south of Tellicherry Arch.

**Comment:** The color scale used in the map is a rainbow scale, which should be avoided (see the work by Fabio Crameri on the use of color in scientific publications). Moreover, the scale has no clear zero value color: a scale that has both positive and negative values should have a clear zero value color to avoid artifacts and apparent structures.
Reply: We thank the editor for the comment. Scientific color scale is now used for all figures in the MS.

**Comment:** I see in this figure that there is an additional zoom-in to the study area. This becomes rather confusing, as there are now two zoom-ins (study areas?) of the general area shown in Fig. 1. It would be good to only use one extent to present the model results for consistency. It is now rather difficult to for instance compare the structures shown in Fig. 2 with those shown in Fig. 3 → are the lows in fact tracing the interpreted grabens? It
- Also, there should be an indication in the caption that the location of this map is shown in Fig. 1.
Reply: The figures are now modified and two study area problem is now resolved.

**Comment:** The sections miss an indication that the seconds are in TWT. At the least, this should be indicated in the caption (including a definition of TWT).
Reply: This is now added in the figure and corrected in the text.

**Comment:** Using circles to indicate circles is a bit confusing→ one may mistake it for a zoom-in. It would probably better to just use an arrow, or perhaps a dotted circle instead.
Reply: This is now represented in the figure as dotted circles.

**Comment:** The white arrow indicating the Tellicherry Arch is poorly visible. Consider using another color. (same for other figures)

Reply: This is now changed in the figure.

**Comment:** Caption: what does "CRS represents the Cannanore Rift System as identified by DGH" mean? What is "DGH" an abbreviation of? Please specify.
Reply: DGH stands for Directorate General of Hydrocarbons. The CRS is identified by DGH, a body under the government of India. They used seismic data to identify the features but the details of the seismic sections are not available in the public domain. This is now described in the text (See discussion part and reference to the report by DGH for more details).

**Comment:** Note that the "broken brown line" is very poorly visible in the map. Please improve this.
- Note that the line is in fact not broken (?)

Reply: The broken line is now boldfaced for better visibility.

**Comment:** The horizontal scale of BB' and CC' is different from those in the other sections (which appear to also represent variable lengths in map view. It may look esthetically pleasing to have these sections in the figure all at the same size, but it does not properly represent the natural situation and the relations between these sections. Please rescale things.
- This is also relevant to Fig. S2

**Comment:** Seismic line labeling: why are some of these lines labels with numbers, and other with letters? Please standardize things.
Reply: The horizontal scale is now rescaled and all the seismic lines are now labelled with alphabets (please see figure 4).

**Comment:** Overall, only faults are interpreted in these seismic sections. Is there no data whatsoever about ages etc.? There is a mention of various boreholes in the area, so I would think this could be added? → like is done for sections 1-3 in the supplement.
- Note that there is various annotation in sections 1-3 that is not explained anywhere (no legend)

Reply: The sections present in 1-3 (now shown in S1) is from different study and not this study. Unfortunately, no age data is available from the wells.

**Comment:** Wy are the grabens in the Trivandrum Terrace area indicated in white? They are barely visible. Please use the same color as used to the west.
- Same for the NW corner of the map

Reply: This was done to distinguish the grabens with different orientation but since it is not visible we have changed made similar to the other grabens marked.

**Comment:** Upon closer inspection of the seismic sections: it seems that there are many faults that were not interpreted. Why not? In fact, I realized that the Laccadive Basin is the study area, but there is not one section that clearly shows the general characteristics of the basin (it is a rifted basin right?) → I would suggest having a look at (the figures of) Gireesh & Pandey (2014) → Open Access link: www.researchgate.net/publication/260213497
Reply: We have now included a seismic section from Unnikrishnan, 2018 to show the general characteristic of the Laccadive Basin and adjoining area. The focus of the study is to understand

the opening of the Laccadive Basin and hence we have not undertaken a detailed seismic interpretation in the area.

**Figure 3**

**Comment:** Panel (E) is described as a tectonic map in the caption, which it is not really? (panel F seems to be?)
- Note that panel F is not a map of beta-values, as described in the caption
- Overall, panels (E) and (F) seem to represent general interpretations, rather than results, and should as such be made into separate discussion figures.

Reply: There was a mistake in the labelling of diagram which is now corrected. The tectonic map, the beta-value map and the depth to basement map are now included as a separate diagram in the MS (fig.7 in the revised MS).

**Comment:** See comments on the use of scientific color (scales) in Fig. 2. These are also relevant here; the color scales in Fig. 3 seem inappropriate.
- Color scale units are not always aligned in the same way (compare panel A with the other panels.

Reply: Scientific color scale is now used in figures.

**Comment:** The lows are indicated using red lines. These lines are poorly visible: please use another indication (e.g., black dotted lines).
- The same for the CKE in green.

Reply: The changes are made in the figure accordingly.

**Comment:** Caption: the abbreviations of TA and TT are nor provided, please add these.

Reply: This is now explained in the caption.

**Comment:** Caption: the repeated "with all identifications" is a bit vague. Consider using "with all identified/interpreted structures"

Reply: We thank the editor for the above comments and the caption is modified accordingly.

**Comment:** In panels B-F, but not in A, there are additional lines in the SE. What do these represent (it is not clear what "shelfal tectonic elements" are, and why there are not indicated in panel A)?

Reply: The thin black lines close to the coast represent major faults identified on the continental shelf (Singh and Lal, 1993).

**Figure 4**

**Comment:** Somehow the study area has a different extent than that in Fig. 3? Please standardize the study area extent in your maps for consistency.

Reply: The extent of the area is now standardized for consistency and easy comparison between different maps. The sediment thickness map is only available from $8^0$ N, hence the area below it is blanked.

**Comment:** See previous comments on the use of colors. This needs to be improved here.

- It would be best to use the same scale for panels A-C, to allow for easy comparison between the different time intervals

Reply: Using the same scale actually makes it difficult to appreciate the map since the sediment deposition in 6B is not comparable with the other time intervals. But we have used scientific color scale for the maps now and same color scale for panels A and C which are comparable.

**Comment:** I suggest using a broken line or something less dominant to indicate the sediment patch in panels A and B.
Reply: The figure is modified accordingly.

**Figure 5**

**Comment:** This figure needs to be larger to better show the details
Reply: The figure is enlarged to better show the details.
**Comment:** Stage III covers no less than 40 Myr, but seems to show a snapshot of the initial Laccadive Basin opening (around 60 Myr?). I strongly suggest avoiding having such time ranges in these panels, as it is confusing.
Reply: We agree with the editor and the time ranges are removed since it creates confusion and we don't have very strong control on the age range of the events.

**Comment:** Stage IV: I would simply remove Madagascar to avoid confusion. The way it is now shown, it seems to suggest India and Madagascar are pretty close to each other, with the black line representing a mid-oceanic ridge.
Reply: We agree with the editor and Madagascar is removed from the figure to avoid confusion.

**Comment:** It would be useful to add some annotation highlighting the important events in the system.
- Note the timing of the various events: how do we know the age of the rifting that is attributed to stage II? This is not really specified/justified in the text?
- See also the comment on the last part of the discussion: it would be good to
Reply: We thank the editor for the above comments. As mentioned in an earlier reply, the extension south of Tellicherry Arch do not correlate with any major structural trends on the onshore region. Further, as mentioned in the text, the Laccadive area was adjacent to the spreading centre in the Mascarene Basin and studies propose (Shuhail et al., 2018) the spreading in the Mascarene Basin was connected to CKE through long transform faults. This prompted us to suggest this timing for the formation of rift system. Later on, due to extension these rifts widened to open the Laccadive Basin.

**Comment:** I suggest moving the text "Stage-I" etc. in each panel to the bottom-right corner (it seems poorly aligned at the moment.
- Also, the header of the Stage-I panel seems not properly aligned
Reply: We thank the editor for the above comment and the changes are made accordingly.

**Comment:** Caption: please provide the meaning of ATTC and CKE (each abbreviation in a caption/figure needs to be explained in the caption [of that figure]).
Reply: We thank the editor for the above comment and the changes are made accordingly.

---

## Editor Decision (ED2)

Dear authors,

Many thanks for submitting a revised version of the manuscript.

I have done a thorough revision, and although this revised version has improved a lot (both text and figures), there are a number of issues that need attention. Please see the uploaded document for more details.

Looking forward to receiving your revised manuscript.

Kind regards,

Frank Zwaan

**General comments:**
- The manuscript is still very short. This can be fine, but it means that things need to be very clear in a very short text. However, in the current manuscript, it seems that important information is missing and descriptions are not always easy to follow. The authors have provided extensive details in their replies, and these details should be available to the reader in some form, if not in the main text, then in the supplement.
- The results section is really too short (17 lines only), followed by a multi-page discussion. This is a bit of a problem. Moreover, the authors should present their results in a logical order (see also below) and avoid presenting new details in the discussion (e.g. detailed description of the distribution of sediments → all information derived from analysis done in this study and that is touched upon in the discussion needs to be clearly laid out in the results section).
    - When reading the methods, it seems that the sedimentary thickness maps are key to prepare the gravity analysis. So, when reading the current text, I would expect the following order of presentation (each with their own sub-section in the results):
        - Sedimentary thickness maps
        - Newly derived Bouguer anomaly maps → revealing key structures
        - Seismic sections → providing more detail on these key structures.
    - Other orders of presentation could also work, e.g. first the seismic sections, then the sedimentary thickness map, then the Bouguer anomaly. Perhaps even the current one could be fine. But the order of things needs to be consistent in both the methods and results to make things easy to follow for the reader, and to set things up for the discussion, where the results are interpreted and combined with current knowledge to develop the tectonic history of the study area.
- The reasons for interpreting one series of grabens being older than the other are not very clearly explained. From geometries and basin distribution alone, this distinction cannot be easily made. The authors should make it very clear what the reasons for their interpretations are. N
    - There are some reasons given here and there, but they are somewhat hidden in the text that seems to already assume that the case is clear. Instead, these reasons need to be highlighted.
- The way the figures are presented is a bit confusing.
    - Firstly, results Fig. 3 and 4 are shown before the results section. This is not a major issue, but needs to be corrected at some point (at the latest in the published paper, if the manuscript is accepted)
    - There is still some ambiguity regarding the study area. There is an "area of interest" shown in Fig. 1b, but a smaller "study area" in later figures. Even so, the authors present seismic sections that are taken from outside the study area (G-G' and H-H'). This needs some fixing. A simple solution would be to state that the "study area" is simply as large as the "area of interest" (in other words, they are the same). The maps in Figs. 5-7 could be made a little larger to accommodate for this I would say. This way, there will be no ambiguity about the extent of the study.
- Some sentences have some grammar issues or are not very clear, I tried to point them out and propose solutions

**Specific comments:**

Line 4: it may be better to use "situated" instead of "lying"

Line 7: "extensional directions" seems a bit confusing, I assume you mean "trends of extensional structures"? This needs some rephrasing

Line 10: "associate" is used twice in this line, perhaps the last occurrence could be "linked"?

Line 11: grabens are by definition extensional features, hence "extensional" should be removed here

Line 18: it should be something like "culminated into the development of the present northwest Indian Ocean" I would say.

Line 18-20: I believe the abbreviations introduced here are not used later in the text? As such, they can be removed here.

Fig. 1: this is very nice figure now. Some comments
- In panel A, the altitude scale is too small (text is not readily readable).
- In panel B, there are numerous abbreviations for the various plates/cratons, which should be specified in the caption
- In Panel B, I would recommend using a bright red dot, instead of a blue dot to indicate the Deccan Volcanic Province
- It may be useful to merge figures 1 and 2 (by simply putting the two panels of Fig. 2 below those of Fig. 1). → see what it may look like on next page
  - Caption text can be merged, saving space.
  - Font size may have to be adjusted in the panels derived from Fig. 2
  - See comment on the order of the panels in Fig. 2

[Figure]

Line 29: it should be "with the eastern Madagascar margin"

Line 29: "based on the matching of the major shear zones and reconstructed to 30 m isobath" seems to be missing some words in the latter part of the sentence. Please rephrase

Line 30-33: "However, recent close-fit reconstruction models have incorporated the continental fragments like Laccadive Ridge (Bhattacharya and Yatheesh, 2015) or Mauritia, comprising of Mauritius, the Southern Mascarene Plateau, the Laccadive Plateau and the Chagos Bank (Torsvik et al., 2013) between India and Madagascar in the India-Madagascar pre-drift scenario, and suggest a breakup timing of around 83 Ma." → it could be better to use:

- "However, recent close-fit reconstruction models have incorporated the continental fragments like **the** Laccadive Ridge (Bhattacharya and Yatheesh, 2015) or Mauritia (comprising of Mauritius, the Southern Mascarene Plateau, the Laccadive Plateau and the Chagos Bank between India and Madagascar in the India-Madagascar pre-drift scenario, Torsvik et al., 2013), and suggest a breakup timing of around 83 Ma.
  - Here the extent of Mauritia is more clearly defined.

Line 34: "Laccadive Ridge (Plateau)" is confusing, as this suggests there are two names (for the same structure?). Please stick to one name and use it consistently through the text. Please also add "The" at the start of this sentence

Line 35: add a figure reference directly after "India"

Line 36: it may have to be "it is well-known" that"

Line 36: the SW margin of what? Please specify in the text.

Line 36: it should be "the end of the Cretaceous"

Fig. 2. Some comments:
- See idea of merging Fig. 2 with Fig. 1
- Consider swapping panels A and B to respect the geographical arrangement shown in Fig. 1a (Madagascar lies to the west of India, so it would only be natural to have Madagascar in Panel A and India in Panel B)
- Panel A: consider filling the well location symbols with a white center to make them stand out
  - Is one of these the CH-1-1 well that is mentioned in the text? Please indicate it (if possible, indicate the names of all wells shown in the figure please
- Panel B: the green on blue lines are poorly visible. Please try black for (old) transform faults and white for the magnetic anomalies
  - Note that it is not very clear which anomaly is which (the annotation indicating the name of the anomalies is not clearly linked to specific anomalies it seems)
- The topography scale is too large, please reduce it a bit in size
- If Fig. 2 will not be merged with Fig. 1, the caption should contain a mention that the location of these maps is shown in Fig 1a.

Line 37: "of wide-spread trap layers" or "of a wide-spread trap layer"

Line 38: please specify to whom the vintage data was not of much help (it now reads a bit as this was not helpful in this manuscript, which is not the case I believe)

Fig. 3. Very nice figure. Some comments:
- Please specify in the caption that panel A shows gravity data, this is not clear
- In both panels: the reddish arrow indicating the Tellicherry Arch is poorly visible. An easy and effective solution is to add a black outline to the arrow to make it stand out.
  - This issue also occurs in various other figures
- The broken red line is very poorly visible in both panels → please use another color (e.g. black)
- The caption should contain a mention that the location of these maps is shown in Fig 1a.

Line 40: see comment on wells in Fig 2 (where is this key well located? Please indicate on a map).

Line 41-43: "One of the key question that was not resolved is the absence of Late Cretaceous sediments in the Laccadive basin as a whole and the long time gap of more than 20 Ma between the India-Madagascar breakup at 83 Ma and the oldest sediments of Paleocene age."

- This sentence is not very clear. I believe it should be something like:
  - "One of the key questions that have not been resolved concerns the absence of Late Cretaceous sediments in the Laccadive basin as a whole: what caused this more than 20 Myr gap in the sedimentary record between India-Madagascar breakup at 83 Ma and the oldest Paleogene sediments?"

Line 45-46: it should be something like "makes **for** a complex geodynamic setting, **considering** how this separation took place, and therefore provides some insights into the **impact of** pre-existing lithospheric inheritance."

- It was not very clear why inheritance is mentioned.

Fig 4: very nice figure, some comments:
- The dotted circles help to show the interpreted intrusions. However, would it be possible to draw in the actual intrusions themselves? Now the reader still needs to identify these intrusions. I think some transparent grey could work well (?)
- Note that the map indicating the locations of the sections has the same issues as Fig. 3a

Line 48: "margin" → perhaps use "Western Continental Margin of India" or WCMI to remind the reader (make it very clear)

Line 48: consider starting a new paragraph at "In this study …" to emphasize that the goal of the current manuscript is being introduced.

Line 49: it should be "at the southwestern part of the margin" to avoid confusion (we are still talking about the WCMI here, not another margin)

Line 51: use "improved **plate** tectonic reconstruction models"

Line 55: the ATTC is never used in the figures (?) → instead "TT" and "AP" is shown → please use a consistent name (I would suggest Trivandrum Terrace/TT (shorter and easier).
- After consideration, perhaps just remove the abbreviation "ATTC" from the main text. Including an abbreviation suggests it is a very important term/feature, which it does not seem to be later in the text (?)

Line 59: "to this" could be removed

Line 61: it may be better to use "DGH 2024" to make it clear this is a citation that can be found in the reference list (now it is only an abbreviation).

Line 63: it should probably be "from the General …"

Line 63-64: these data are shown in Fig. 3 I believe? A reference to this figure would be needed.

Line 64-66: it is not clear whether a new seismic analysis is performed, or whether the authors simply adopted the data from Unnikrishnan et al. (2023). This needs to be made very clear. If the authors did a new analysis, they should explain how the analysis was done (in the main manuscript, or otherwise in the Supplement).

Line 66: double "and" → please remove

Line 74: it should be "for the water column are used, respectively" I believe

Fig. 5: Very nice figure, but same issues as Fig. 3a:

- "Tellicherry arrow" needs a black outline
- The brown, green and blue colors used to indicate the lows, CKE and volcanic ridge are often poorly visible. Please use black instead
- Please check the alignment of the color scale and annotation, it seems like there is some overlap with other elements of the figure

Line 83-85: it can be rephrased: "We correlated these structures with the gravity anomaly trends and noticed that the grabens are oriented NNW-SSE in the area north of the Tellicherry Arch, whereas the grabens are oriented NNE-SSW south of Tellicherry Arch."

Line 85-86: it should be "anomalies, the continuity of which"

Line 91: it should be "with much less sedimentation" I believe (or "with limited sedimentation")

Line 94: "is" should be "was"

Line 96: it should be "to recent times" and "sedimentation has been uniform"

Fig. 6: some comments:
- See previous comments on the "Tellicherry arrows"
- Is this a new results map, or rather a

Line 101-103: very important to get this right, as this is a key point. The text should be something like "This study identifies two major extensional events in the southern part of WCMI, **the first being recorded by** the NNW-SSE oriented grabens over the Laccadive Ridge north of Tellicherry Arch, **and the second by the** NNE-SSW graben system in the Laccadive Basin area south of Tellicherry Arch (fig. 3).
- It is (still) not very clear from the text why the NNW-SSE grabens should be the older structures.

Line 104 and 105: see previous comment on DGH citation

Line 105: the acronym for the directorate is already specified earlier in the text, so why not just use "DGH here"?

Line 104-105: the citation of Zwaan et al. 2021 may be misleading, they did not work on this study area.
- The authors need to indicate the Dharwar structure on a map (it is not clear what this trend is, and where it can be found)
- If they intend to cite the Zwaan et al. study, they need to specify that (and why) the structures in the present study area can be compared with the analogue modelling results by Zwaan et al. (2021).

Line 105-109: the identification of the CRS seems to be a result, and should be included in the results section, before it is introduced in the discussion. (it needs to be clear what the new results are, and what is interpretation and discussion).

Line 112-114: It is not fully clear to me how this conclusion can be drawn. What is the evidence for the age of these grabens? Is there some previous work that provides age constraints? After all, all extensional structures shown (both north and south of the Tellicherry Arch, but also on the Trivandrum Terrace could form in a roughly E-W extensional system). This needs some clarification in the text.

Line 114-119: I do not follow what is meant here. What trend is expected to continue southward? What is different? I assume the orientations of the grabens? Even so, the text needs some clarification.

Fig. 7: some comments:
- Panel A: See comments on color use in for instance Fig. 3a.
- The seismic section should have its own labeling (D)

- Caption: please specify what "LB" stands for

Line 156: please add a reference to Fig. 8 here.

Line 159-161: what is meant with a large number of sutures? Sutures are major boundaries between tectonic plates, so there should generally be a single one. Perhaps the authors mean inherited structures or so? Please rephrase.

Fig. 8: some comments:
- Please make sure to use the same font in all panels (the "Stage" indications are in Times New Roman, the rest in Arial it seems).
- It would be good to add arrows indicating the direction of plate motion in each panel (not 100% clear at the moment)
- Please make sure to (re)align the annotation, it should probably be a bit smaller and perhaps not bold. NB: headers seem fine
- Consider removing (at least) the outer box, as it distracts from the actual figure.
- Stage II: "spreading" is misspelled

Line 164-165: it would be better to put the Peron-Pinvidic etc. references at the end of this sentence, and to specify that these are more general studies/studies focusing on other areas (which could still be used for interpretation here).

Line 168: see previous comment on the Dharwarian trend → it is not made clear what this is

Line 170: what spreading center? I assume the one in the Mascarene basin? Please specify

Line 170-171: where was it stated that the lithosphere was weak? Please check.

Line 174-176: in fact, this seems to not be show in section 5.1 (?) please rework

Line 176-178: what is meant here? That magma from the Réunion (hotspot) magma intruded into transform faults? Please clarify in the text.

Line 176: the Réunion hotspot?

Line 178: what are "similar arguments"? → please specify in the text

---

## Author Response (AR3)

**Reply to editors' comments:**

We thank the editor for going through the manuscript meticulously and providing constructive remarks and comments. We believe that this helped a lot in improving this manuscript.

**General comments:**

**Comment:** The manuscript is still very short. This can be fine, but it means that things need to be very clear in a very short text. However, in the current manuscript, it seems that important information is missing and descriptions are not always easy to follow. The authors have provided extensive details in their replies, and these details should be available to the reader in some form, if not in the main text, then in the supplement.

Reply: We thank the editor for the comment and we have made corrections and included descriptions wherever necessary as suggested by the editor.

**Comment:** The results section is really too short (17 lines only), followed by a multi-page discussion. This is a bit of a problem. Moreover, the authors should present their results in a logical order (see also below) and avoid presenting new details in the discussion (e.g. detailed description of the distribution of sediments → all information derived from analysis done in this study and that is touched upon in the discussion needs to be clearly laid out in the results section).

- When reading the methods, it seems that the sedimentary thickness maps are key to prepare the gravity analysis. So, when reading the current text, I would expect the following order of presentation (each with their own sub-section in the results):
  - Sedimentary thickness maps
  - Newly derived Bouguer anomaly maps → revealing key structures
  - Seismic sections → providing more detail on these key structures.
- Other orders of presentation could also work, e.g. first the seismic sections, then the sedimentary thickness map, then the Bouguer anomaly. Perhaps even the current one could be fine. But the order of things needs to be consistent in both the methods and results to make things easy to follow for the reader, and to set things up for the discussion, where the results are interpreted and combined with current knowledge to develop the tectonic history of the study area.

Reply: We thank the editor for pointing out this. We have tried that to present clean observations in results section as suggested in earlier revisions and later on we have synthesized the discussion by integrating the results from this study and in light of existing understanding of the study area. We have rearranged the results section as suggested.

**Comment:** The reasons for interpreting one series of grabens being older than the other are not very clearly explained. From geometries and basin distribution alone, this distinction cannot be easily made. The authors should make it very clear what the reasons for their interpretations are. N

- There are some reasons given here and there, but they are somewhat hidden in the text that seems to already assume that the case is clear. Instead, these reasons need to be highlighted.

Reply: We agree with the editor and more description is now included to make the case clear. This is now made clear in the first paragraph of discussion section in the revised MS.

**Comment:** The way the figures are presented is a bit confusing.
- o Firstly, results Fig. 3 and 4 are shown before the results section. This is not a major issue, but needs to be corrected at some point (at the latest in the published paper, if the manuscript is accepted)
- o There is still some ambiguity regarding the study area. There is an "area of interest" shown in Fig. 1b, but a smaller "study area" in later figures. Even so, the authors present seismic sections that are taken from outside the study area (G-G' and H-H'). This needs some fixing. A simple solution would be to state that the "study area" is simply as large as the "area of interest" (in other words, they are the same). The maps in Figs. 5-7 could be made a little larger to accommodate for this I would say. This way, there will be no ambiguity about the extent of the study.

Reply: We thank the editor for pointing out this. Now the figures are rearranged accordingly. Regarding the study area, we prefer the maintain the current structure as we would like to focus on the opening of the Laccadive Basin and adjoining area south of Tellicherry Arch in the manuscript. The larger area is shown to give a regional picture of the study area and to show how the orientation of grabens change when coming into the Laccadive Basin.

**Comment:** Some sentences have some grammar issues or are not very clear, I tried to point them out and propose solutions

Reply: We are grateful to the editor for going through the MS in detail and providing with these suggestions.

**Specific comments:**

**Comment:** Line 4: it may be better to use "situated" instead of "lying"
Reply: We thank the editor for the comment. The suggestion is included.

**Comment:** Line 7: "extensional directions" seems a bit confusing, I assume you mean "trends of extensional structures"? This needs some rephrasing
Reply: The suggestion is included.

**Comment:** Line 10: "associate" is used twice in this line, perhaps the last occurrence could be "linked"?

Reply: The text is modified accordingly.

**Comment:** Line 11: grabens are by definition extensional features, hence "extensional" should be removed here
Reply: We thank the editor for the comment. The text is modified accordingly.

**Comment:** Line 18: it should be something like "culminated into the development of the present northwest Indian Ocean" I would say.

Reply: The text is modified accordingly.

**Comment:** Line 18-20: I believe the abbreviations introduced here are not used later in the text?

As such, they can be removed here.

Reply: We thank the editor for pointing out this. The is now removed.
Comment: Line 29: it should be "with the eastern Madagascar margin"

Reply: The text is modified accordingly.

Comment: Line 29: "based on the matching of the major shear zones and reconstructed to 1000 m isobath" seems to be missing some words in the latter part of the sentence. Please rephrase

Reply: We thank the editor for the comment. Katz and Premoli 1979 explained the fit of India and Madagascar based on matching shear zones onland India and Madagascar and the coastlines fitted to 1000 m isobath. It was mistakenly written as 2000 m isobath which is corrected now and the sentence is modified for clarity.

Comment: Line 30-33: "However, recent close-fit reconstruction models have incorporated the continental fragments like Laccadive Ridge (Bhattacharya and Yatheesh, 2015) or Mauritia, comprising of Mauritius, the Southern Mascarene Plateau, the Laccadive Plateau and the Chagos Bank (Torsvik et al., 2013) between India and Madagascar in the India-Madagascar pre-drift scenario, and suggest a breakup timing of around 83 Ma." → it could be better to use:
- "However, recent close-fit reconstruction models have incorporated the continental fragments like **the** Laccadive Ridge (Bhattacharya and Yatheesh, 2015) or Mauritia (comprising of Mauritius, the Southern Mascarene Plateau, the Laccadive Plateau and the Chagos Bank between India and Madagascar in the India- Madagascar pre-drift scenario, Torsvik et al., 2013), and suggest a breakup timing of around 83 Ma.
- Here the extent of Mauritia is more clearly defined.

Reply: We thank the editor for the comment. Now the text is modified as *"However, recent close-fit reconstruction models have incorporated the continental fragments like the Laccadive Ridge (Bhattacharya and Yatheesh, 2015) or Mauritia (comprising of Mauritius, the Southern Mascarene Plateau, the Laccadive Plateau and the Chagos Bank) (Torsvik et al., 2013) between India and Madagascar in the India-Madagascar pre-drift scenario, and suggest a breakup timing of around 83 Ma."*

Comment: Line 34: "Laccadive Ridge (Plateau)" is confusing, as this suggests there are two names (for the same structure?). Please stick to one name and use it consistently through the text. Please also add "The" at the start of this sentence

Reply: The text is modified accordingly.

Comment: Line 35: add a figure reference directly after "India"

Reply: The suggestion is incorporated.

Comment: Line 36: it may have to be "it is well-known" that"

Reply: The text is modified accordingly.

Comment: Line 36: the SW margin of what? Please specify in the text.

Reply: The text is modified for clarity as per the suggestion.

**Comment:** Line 36: it should be "the end of the Cretaceous"

Reply: The text is modified accordingly.

**Comment:** Line 37: "of wide-spread trap layers" or "of a wide-spread trap layer"

Reply: The text is modified accordingly.

**Comment:** Line 38: please specify to whom the vintage data was not of much help (it now reads a bit as this was not helpful in this manuscript, which is not the case I believe)

Reply: We thank the editor for pointing out this and this sentence is reframed for clarity.

**Comment:** Line 40: see comment on wells in Fig 2 (where is this key well located? Please indicate on a map).

Reply: We thank the editor for the suggestion and the location of the key well CH-1-1 is now included in the diagram.

**Comment:** Line 41-43: "One of the key question that was not resolved is the absence of Late Cretaceous sediments in the Laccadive basin as a whole and the long time gap of more than 20 Ma between the India-Madagascar breakup at 83 Ma and the oldest sediments of Paleocene age."
- This sentence is not very clear. I believe it should be something like:
  - "One of the key questions that have not been resolved concerns the absence of Late Cretaceous sediments in the Laccadive basin as a whole: what caused this more than 20 Myr gap in the sedimentary record between India-Madagascar breakup at 83 Ma and the oldest Paleogene sediments?"

Reply: The text is modified as per the suggestion of the editor.

**Comment:** Line 45-46: it should be something like "makes **for** a complex geodynamic setting, **considering** how this separation took place, and therefore provides some insights into the **impact of** pre-existing lithospheric inheritance."
- It was not very clear why inheritance is mentioned.

Reply: The text is modified accordingly. We believe and the data suggest that the inheritance has impacted the formation and orientation of grabens in the study area. Hence, we have mentioned inheritance here.

**Comment:** Line 48: "margin" → perhaps use "Western Continental Margin of India" or WCMI to remind the reader (make it very clear)

Reply: We thank the editor for the suggestion and the text is modified accordingly.

**Comment:** Line 48: consider starting a new paragraph at "In this study …" to emphasize that the goal of the current manuscript is being introduced.

Reply: We agree with the editor and the text is modified accordingly.

**Comment:** Line 49: it should be "at the southwestern part of the margin" to avoid confusion (we are still talking about the WCMI here, not another margin)

Reply: We agree with the editor and the text is modified accordingly.

**Comment:** Line 51: use "improved **plate** tectonic reconstruction models"

Reply: The text is modified accordingly.

**Comment:** Line 55: the ATTC is never used in the figures (?) → instead "TT" and "AP" is shown ☐ please use a consistent name (I would suggest Trivandrum Terrace/TT (shorter and easier).
- After consideration, perhaps just remove the abbreviation "ATTC" from the main text. Including an abbreviation suggests it is a very important term/feature, which it does not seem to be later in the text (?)

Reply: We thank the editor for pointing out this. The abbreviation is now removed from the MS.

**Comment:** Line 59: "to this" could be removed

Reply: The text is modified accordingly.

**Comment:** Line 61: it may be better to use "DGH 2024" to make it clear this is a citation that can be found in the reference list (now it is only an abbreviation).

Reply: The text is modified accordingly.

**Comment:** Line 63: it should probably be "from the General …"

Reply: The text is modified accordingly.

**Comment:** Line 63-64: these data are shown in Fig. 3 I believe? A reference to this figure would be needed.

Reply: The text is modified accordingly.

**Comment:** Line 64-66: it is not clear whether a new seismic analysis is performed, or whether the authors simply adopted the data from Unnikrishnan et al. (2023). This needs to be made very clear. If the authors did a new analysis, they should explain how the analysis was done (in the main manuscript, or otherwise in the Supplement).

Reply: We thank the editor for asking this. New seismic data interpretation was carried out in the study whereas sediment isochron maps were directly adopted from Unnikrishnan et al., 2023 and interpreted on the basis of inputs from seismic interpretation and gravity data. This is now made clear in the MS.

**Comment:** Line 66: double "and" → please remove

Reply: The text is modified accordingly.

**Comment:** Line 74: it should be "for the water column are used, respectively" I believe

Reply: The text is modified accordingly.

**Comment:** Line 83-85: it can be rephrased: "We correlated these structures with the gravity anomaly trends and noticed that the grabens are oriented NNW-SSE in the area north of the Tellicherry Arch, whereas the grabens are oriented NNE- SSW south of Tellicherry Arch."

Reply: We thank the editor for the comment. The text is modified accordingly.

**Comment:** Line 85-86: it should be "anomalies, the continuity of which"

Reply: The text is modified accordingly.

**Comment:** Line 91: it should be "with much less sedimentation" I believe (or "with limited sedimentation")

Reply: The text is modified accordingly.

**Comment:** Line 94: "is" should be "was"

Reply: The text is modified accordingly.
**Comment:** Line 96: it should be "to recent times" and "sedimentation has been uniform"

Reply: The text is modified accordingly.

**Comment:** Line 101-103: very important to get this right, as this is a key point. The text should be something like "This study identifies two major extensional events in the southern part of WCMI, **the first being recorded by** the NNW-SSE oriented grabens over the Laccadive Ridge north of Tellicherry Arch, **and the second by the** NNE-SSW graben system in the Laccadive Basin area south of Tellicherry Arch (fig. 3).
- It is (still) not very clear from the text why the NNW-SSE grabens should be the older structures.

Reply: We thank the editor for the comment and the suggestion is incorporated in the MS. More arguments are now included in the text regarding the relative age of the extensional features.

**Comment:** Line 104 and 105: see previous comment on DGH citation.

Reply: The text is modified accordingly.

**Comment:** Line 105: the acronym for the directorate is already specified earlier in the text, so why not just use "DGH here"?

Reply: The text is modified accordingly.

**Comment:** Line 104-105: the citation of Zwaan et al. 2021 may be misleading, they did not work on this study area.
- The authors need to indicate the Dharwar structure on a map (it is not clear what this trend is, and where it can be found)

- If they intend to cite the Zwaan et al. study, they need to specify that (and why) the structures in the present study area can be compared with the analogue modelling results by Zwaan et al. (2021).

Reply: Dharwar trend in now marked in figure 1D. We agree to the comment and the reference to the study of Zwaan et al., 2021 is removed.

**Comment:** Line 105-109: the identification of the CRS seems to be a result, and should be included in the results section, before it is introduced in the discussion. (it needs to be clear what the new results are, and what is interpretation and discussion).

Reply: The DGH identified the outline and orientation of the CRS on the Laccadive ridge through the analysis of large set of unpublished seismic data. The rift identifications and its orientation from this study fall within the outline of CRS as identified by DGH (source: DGH2024). The objective of the study is to examine the seismic lines in this region is to understand the difference in rift orientation north and south of Tellicherry Arch in context of Laccadive Basin formation. Hence, we only presented the rift identifications in the results section.

**Comment:** Line 112-114: It is not fully clear to me how this conclusion can be drawn. What is the evidence for the age of these grabens? Is there some previous work that provides age constraints? After all, all extensional structures shown (both north and south of the Tellicherry Arch, but also on the Trivandrum Terrace could form in a roughly E-W extensional system). This needs some clarification in the text.

Reply: We thank the editor for pointing out the Lack of clarity in this part. Unfortunately, there is no age constrain to draw this conclusion regarding the relative age of the extensional features. But we believe there are some factors to be considered which help to get some idea about the timing of the extensional features. Firstly, the Dharwar trend is observed in the shelf region (Singh and Lal, 1993) and further offshore (Kolla & Cumes, 1990). The initial spreading in the Mascarene Basin was E-W which is orthogonal to the Precambrian Dharwar trend and hence we infer that the NNW-SSE oriented grabens formed during this period. Secondly, the study area was next to the spreading centre in the Mascarene Basin and the spreading was characterized by large transform faults. The spreading was more active in the southern Mascarene basin which was next to southern part of the margin (south of Tellicherry Arch). This evidence comes from the magnetic anomalies in the Mascarene basin and Shuhail et al., 2018 proposed that the spreading was connected to CKE through a long transform fault. Hence, we prefer to interpret the development of the NNE-SSE oriented rift system during this time. This is now clarified in the discussion part.

**Comment:** Line 114-119: I do not follow what is meant here. What trend is expected to continue southward? What is different? I assume the orientations of the grabens? Even so, the text needs some clarification.

Reply: Yes, we meant the orientation of grabens. The text is modified for clarity.

**Comment:** Line 156: please add a reference to Fig. 8 here.

Reply: The reference is now added.

**Comment:** Line 159-161: what is meant with a large number of sutures? Sutures are major

boundaries between tectonic plates, so there should generally be a single one. Perhaps the authors mean inherited structures or so? Please rephrase.

Reply: We thank the editor for pointing out this. We mistakenly mentioned suture zone instead of shear zones. This is now corrected in the MS.

**Comment:** Line 164-165: it would be better to put the Peron-Pinvidic etc. references at the end of this sentence, and to specify that these are more general studies/studies focusing on other areas (which could still be used for interpretation here).

Reply: The text is modified accordingly.

**Comment:** Line 168: see previous comment on the Dharwarian trend → it is not made clear what this is

Reply: We thank the editor for asking this. Dharwar trend is now marked in figure 1C and is the trend of the Precambrian Dharwar Craton (DC) on the onshore.

**Comment:** Line 170: what spreading center? I assume the one in the Mascarene basin? Please specify

Reply: Yes, we meant the spreading in the Mascarene basin. This is now clarified in the text.

**Comment:** Line 170-171: where was it stated that the lithosphere was weak? Please check.

Reply: We meant to say that lithosphere had zones of weakness as evidence by the presence of number of shear zones. We have clarified this in the text now.

**Comment:** Line 174-176: in fact, this seems to not be show in section 5.1 (?) please rework
**Comment:** Line 176-178: what is meant here? That magma from the Réunion (hotspot) magma intruded into transform faults? Please clarify in the text.
**Comment:** Line 176: the Réunion hotspot?
**Comment:** Line 178: what are "similar arguments"? → please specify in the text

Reply: We thank the editor for pointing out the mistakes and lack of clarity through the above comments. We have rewritten the part for more clarity as follows. *"As discussed in section 5.2 distribution of bathymetric highs and intrusives south of Tellicherry Arch provide some evidence for this. It is very likely that, later when the Réunion plume passed over the area, magma may have migrated through the faults formed during this stage giving rise to the preferred orientation of intrusive and bathymetric features in this area. It is worthwhile to note that Bijesh et al. (2018) related the genesis of the bathymetric features to hotspot volcanism"*

**Comment on figures:**

**Comment:** Fig. 1: this is very nice figure now. Some comments
  - In panel A, the altitude scale is too small (text is not readily readable).
  - In panel B, there are numerous abbreviations for the various plates/cratons, which should be specified in the caption
  - In Panel B, I would recommend using a bright red dot, instead of a blue dot to indicate the Deccan Volcanic Province

- It may be useful to merge figures 1 and 2 (by simply putting the two panels of Fig. 2 below those of Fig. 1).
  - → see what it may look like on next page
    - ○ Caption text can be merged, saving space.
    - ○ Font size may have to be adjusted in the panels derived from Fig. 2
    - ○ See comment on the order of the panels in Fig. 2

**Comment:** Fig. 2. Some comments:
- See idea of merging Fig. 2 with Fig. 1
- Consider swapping panels A and B to respect the geographical arrangement shown in Fig. 1a (Madagascar lies to the west of India, so it would only be natural to have Madagascar in Panel A and India in Panel B)
- Panel A: consider filling the well location symbols with a white center to make them stand out
  - ○ Is one of these the CH-1-1 well that is mentioned in the text? Please indicate it (if possible, indicate the names of all wells shown in the figure please
- Panel B: the green on blue lines are poorly visible. Please try black for (old) transform faults and white for the magnetic anomalies
  - ○ Note that it is not very clear which anomaly is which (the annotation indicating the name of the anomalies is not clearly linked to specific anomalies it seems)
- The topography scale is too large, please reduce it a bit in size
- If Fig. 2 will not be merged with Fig. 1, the caption should contain a mention that the location of these maps is shown in Fig 1a.

Reply to comment on figure 1& 2: We thank the editor for the comment. Figure 1&2 are merged and the panels rearranged as suggest by the editor. All changes are made as per the suggestion. The magnetic anomalies in the Mascarene basin is are reproduced from Bhattacharya and Yatheesh 2015 & Shuhail et al., 2018. We suggest the readers to go through the references for more details.

**Comment:** Fig. 3. Very nice figure. Some comments:
- Please specify in the caption that panel A shows gravity data, this is not clear
- In both panels: the reddish arrow indicating the Tellicherry Arch is poorly visible. An easy and effective solution is to add a black outline to the arrow to make it stand out.
  - ○ This issue also occurs in various other figures
- The broken red line is very poorly visible in both panels → please use another color (e.g. black)
- The caption should contain a mention that the location of these maps is shown in Fig 1a.

Reply to comment: All suggestions are incorporated in the revised figure.

**Comment:** Fig 4: very nice figure, some comments:
- The dotted circles help to show the interpreted intrusions. However, would it be possible to draw in the actual intrusions themselves? Now the reader still needs to identify these intrusions. I think some transparent grey could work well (?)
- Note that the map indicating the locations of the sections has the same issues as Fig. 3a

Reply to comment: All suggestions are incorporated in the revised figure.

**Comment:** Fig. 5: Very nice figure, but same issues as Fig. 3a: "Tellicherry arrow" needs a black outline

- The brown, green and blue colors used to indicate the lows, CKE and volcanic ridge are often poorly visible. Please use black instead
- Please check the alignment of the color scale and annotation, it seems like there is some overlap with other elements of the figure

Reply to comment: All suggestions are incorporated in the revised figure.

**Comment:** Fig. 6: some comments:

- See previous comments on the "Tellicherry arrows"
- Is this a new results map, or rather a

Reply to comment: This map is taken from Unnikrishnan et al., 2023 (which is a clean map without any interpretation.) and interpreted incorporating inputs from the seismic interpretation and analysis of gravity maps. All suggestions are incorporated in the revised figure.

**Comment:** Fig. 7: some comments:

- Panel A: See comments on color use in for instance Fig. 3a.
- The seismic section should have its own labeling (D)
- Caption: please specify what "LB" stands for

Reply to comment: We thank the editor for the comment. All suggestions are incorporated in the revised figure.

**Comment:** Fig. 8: some comments:

- Please make sure to use the same font in all panels (the "Stage" indications are in Times New Roman, the rest in Arial it seems).
- It would be good to add arrows indicating the direction of plate motion in each panel (not 100% clear at the moment)
- Please make sure to (re)align the annotation, it should probably be a bit smaller and perhaps not bold. NB: headers seem fine
- Consider removing (at least) the outer box, as it distracts from the actual figure.
- Stage II: "spreading" is misspelled

Reply to comment: We thank the editor for the comment and pointing out the mistakes. All suggestions are incorporated in the revised figure.

---

## Author Response (AR4)

**Reply to Major Comments:**

**Comment:** - there is a description of present-day structural orientations and extension directions. However, it is clear that these may not represent the situation in the past, as plates have moved around (this is very clear from their own summary Fig. 7). As such, one should refrain from stating things like a E-W extension during the opening of the Mascarene basin, that is indicated by the magnetic anomalies, as the situation was different.

Reply: We thank the reviewer for asking this. We believe there is some confusion regarding the comment. First of all, the plate-tectonic reconstruction study carried out by Shuhail et al., 2018 and Bhattacharya and Yatheesh 2015 are based on a fixed Madagascar frame and it is the convention to describe the features based on present-day structural orientations. Secondly, the magnetic anomalies in the Mascarene Basin is well preserved. Moreover, when we analyze the plate movements from reconstruction we observe clearly that the Madagascar and India separation took place E-W initially. Hence, we stated that E-W extension during India-Madagascar separation.

**Comment:** - there is now an argumentation for a change in extensional regime, but it is not really clear how this works. The current text mentions transform faults, but how would such a transform fault change the (local) extension direction I don't see. If anything, the arrangement of spreading centres as shown in stage II of Fig. 7 would imply that extension is roughly SW-NE, so the proposed second-phase grabens are oriented in the wrong direction? Or is the idea that these grabens follow the orientation of the transform faults? if so, this should be made clearer (even so, one would expect the same thing to happen to the north as well). Overall, the evidence provided for stage II is not sufficiently convincing as is (in fact, the text at the start of the discussion seems to imply that both graben orientations in the area developed at the same time), and the mechanism that would cause the development of a later set of grabens with a different orientation remains unclear.

Reply: We thank the reviewer for pointing out this. Yes, the idea is that the grabens follow the orientation of the transform faults. We agree there is some lack of clarity in the text. We have now rewritten the second paragraph of discussion to clarify this. Also, please see the reply to line by line comments.

**Reply to line by line comments in the text:**

**Comment:** write: "India and Madagascar that created the Mascarene Basin in the Late Cretaceous". (otherwise it is not very clear what the Mascarene Basin, which is mentioned later on, really is). **[line 2]**
Reply: The suggestion is incorporated in the MS.

**Comment:** write: "a NNW-SSE oriented structure over the Laccadive Ridge north of Tellicherry Arch, interpreted to result from ENE-WSW extension, and a SSW-NNE oriented structure in the

Laccadive Basin region towards the south, interpreted to result from NW-SE extension." Otherwise, the text does not make much sense **[line 7-8]**

Reply: We thank the editor for pointing this out. The suggestion is incorporated in the MS.

**Comment:** write: "Paleocene trap volcanics" to make clear we are talking about volcanics. **[line 10]**

Reply: The suggestion is incorporated in the MS.

**Comments:** (fig.1 caption)

- please check the ages of break-up in this map and this citation, as they seem to be incorrect. As far as I am aware, break-up in the North Atlantic did not occur prior to Jurassic, but here it is indicated as Triassic?
- write: "Map showing"
- write: "of the maps in panels C and D are shown in panel A"

Reply: We thank the reviewer for pointing out this mistake. We have now corrected this in figure and checked for the correctness of other times. The other suggestions are incorporated as per the comment.

**Comment:** please indicate the Laxmi Ridge on one of the maps, --> it is now missing and therefore unclear /confusing **[line 25]**

Reply: We have now indicated Laxmi Ridge in fig 1A.

**Comment:** 2 times "matching" --> consider using a synonym? **[line 29]**

Reply: The suggestion is incorporated in the MS. The sentence is modified as "The southern part of the margin is considered to be conjugate with the eastern Madagascar margin (Katz and Premoli, 1979) based on the continuity of the major shear zones and coastlines matched at 1000 m isobath."

**Comment:** move figure reference to end of sentence for clarity **[line 35]**

Reply: The suggestion is incorporated in the MS.

**Comment:** write "a wide-spread layer of trap volcanics" **[line 36-37]**

Reply: The suggestion is incorporated in the MS.

**Comment:** start the sentence with "By contrast, the" **[line 40]**

Reply: The suggestion is incorporated in the MS.

**Comment:** add reference to Fig. 1D here **[line 40]**

Reply: The suggestion is incorporated in the MS.

**Comment:** use "felsic volcanics"?  NB: this is basement, right? if so, write: "felsic volcanics that are attributed to the basement" or simply "felsic volcanics of the basement" **[line 41]**

Reply: These volcanics are assumed to be Late Cretaceous in age and acidic (> 65 % of SiO2) in chemical composition (Singh and Lal, 1993; Rathore et al., 2015). The basement in this region is supposed to be Dharwarian supergroup rocks (Archean) which is yet to be established by drilling. The CH-1-1 well was terminated in acidic volcanics which are correlated with Late Cretaceous volcanism of St. Mary's Island (Pande et al., 2001).

**Comment:** PROBLEM? **[line 44-45]**

Reply: The purpose of the statement is to imply that we are trying to address this issue and the opening of the Laccadive Basin. Our model of opening of Laccadive Basin can help to address this issue.

**Comment:** remove hyphen **[line 44]**

Reply: The suggestion is incorporated in the MS.

**Comment:** replace with "presence" (makes more sense in this context) **[line 45]**

Reply: The suggestion is incorporated in the MS.

**Comment:** write "WCMI" **[line 49]**

Reply: The suggestion is incorporated in the MS.

**Comment:** put between parentheses **[line 57]**

Reply: The suggestion is incorporated in the MS.

**Comment:** this should be "Fig. 2A" **[line 66]**

Reply: The suggestion is incorporated in the MS.

**Comment:** please use "Figs. 2b-5" to avoid citing later figures earlier than early figures (in this case, Fig. 5 is cited before Figs. 3 and 4). **[line 68]**

Reply: The suggestion is incorporated in the MS.

**Comment:** write: "to obtain" **[line 72]**

Reply: The suggestion is incorporated in the MS.

**Comment:** detail: Solid Earth will likely convert this in kg/m3 --> you could already do so yourself **[line 76-77]**

Reply: The suggestion is incorporated in the MS.

**Comment:** add the source of the gravity data (Sandwell et al. 2014) **[fig.2 caption]**

Reply: We thank the reviewer for the comment. The suggestion is incorporated in the MS.

**Comment:** this is not really a new result? (the data are simply taken from Unnikrishnan et al. 2023). To avoid confusion, please use the following title: "Sediment Isochron map analysis" that way, it's clearer that this is not really a new map per sé (what seems to be implied with the current title), but that a new analysis is done with it. **[line 86]**

Reply: The suggestion is incorporated in the MS.

**Comment:** write "below" **[line 92]**

Reply: The suggestion is incorporated in the MS.

**Comment:** write: "From Early Miocene to recent times, sedimentation" **[line 93]**

Reply: The suggestion is incorporated in the MS.

**Comment:** make it very clear that this is directly based on the data from Unnikrishnan et al 2023. Omission of this fact is not acceptable here. **[fig. 3 Caption]**

Reply: The suggestion is incorporated in the MS.

**Comment:** similar to the previous section, it would be good to imply that something new is done --> replace "map" with "mapping"? **[line 97]**

Reply: The suggestion is incorporated in the MS.

**Comment:** write: "reveal" **[line 98]**

Reply: The suggestion is incorporated in the MS.

**Comment:** refer to Fig. 2B **[line 99]**

Reply: The suggestion is incorporated in the MS.

**Comment:** write: "structural trends" **[line 113]**

Reply: The suggestion is incorporated in the MS.

**Comment:** move figure reference to end of sentence **[line 116]**

Reply: The suggestion is incorporated in the MS.

**Comment:**

this is what the current-day situation shows, but was this also the case in the past (the plates may have moved over time, so these spreading directions may in fact not be correct for the past situation?) **[line 121]**

see comments on the validity of plate motion directions presented in this manuscript **[line 130]**

Reply: We thank the reviewer for asking these. This is a convention that is followed when plate-tectonic reconstruction studies are carried out. The directions are mentioned based on the present-day configuration. Moreover, the reconstruction study indicates the sense of movement which matches with the conclusions drawn correctly.

**Comment:** see general comment: lines 125-131 do not develop a convincing argument yet. Please rework this and make things clearer. A reference to a figure would be of great help as well, otherwise it is not very clear what the tectonic setting is, that the authors invoke. Fig. 7 seems to be the obvious figure to refer to here. **[line 125]**

Reply: We thank the editor for pointing out this. We have now rewritten the part for more clarity.

**Comment:** write: "reconstructions" (plural), or "reconstruction efforts" **[line 128]**

Reply: The suggestion is incorporated in the MS.

**Comment:** not clear how this connection is made. Also, transform faults are in principle only active between the two spreading ridges they connect, so I don't see how they can affect grabens far away on the margin of India? **[line 129]**

Reply: We agree with the reviewer that transform faults are in principle only active between two spreading ridges. The southern part of the Mascarene Basin was characterized by large transform faults and the spreading was very active here compared to the north. And during this time the study area was very close to this spreading centre. Hence, we suggest in our model that, this proximity lead to formation of faults in the region with same orientation as that of the transform faults.

**Comment:** The text seems to suggest this is all still happening at the same time as the opening of the Mascarene Basin? (i.e. at the same time as the opening of the other grabens to the north?) --> so there is no phase II, in fact? **[line 129]**

Reply: No, we did not indent to suggest that. The paragraph is reframed for clarity.

**Comment:** what isochron maps? please provide citations **[line 131]**

Reply: We thank the editor for the comment. The reference to figure 3 is now added here.

**Comment:** write: "extensional deformation in rifts and rifted margin systems" **[line 172]**

Reply: The suggestion is incorporated in the MS.

**Comment:** the spreading ridges have different colors, please specify what this means. Also, it would be good to differentiate between spreading ridges and transform faults so that the reader will not mix them up (this is especially confusing in stage II, where extension could be interpreted both as SW-NE and NW-SE). Alternatively, add some arrows to indicate general plate motion directions **[fig. 7 caption]**

Reply: We thank the editor for the suggestion and the diagram is modified accordingly. Arrows are shown along the spreading centres to distinguish it from transform faults and show the direction of spreading. Extinct spreading centre is now marked in the diagram.

**Comment:** this is what is supposed to happen in stage III, so can't be of interest in stage II, it seems to me. **[line 192]**

Reply: We agree with the reviewer and this sentence is removed from stage II and mentioned in the second paragraph of discussion for clarity.

**Comment:** write "basin in the Paleocene" **[line 199]**

Reply: The suggestion is incorporated in the MS.

---

## Author Response (AR5)

**Reply to editor's comments:**

We thank the editor for going through the manuscript thoroughly and suggesting corrections. All technical corrections are incorporated as suggested by the editor.

**Comment:** Line 134: "Mascarene Basin spreading center" (please also add a reference to Fig. 7C/stage III to help the reader understand the tectonic context, which will otherwise remain unclear)

Reply: We agree with the editor and the changes are incorporated as suggested by the editor.